# Targeting ribosome biogenesis as a novel therapeutic approach to overcome EMT-related chemoresistance in breast cancer

Yi Ban[1,2]*[†], Yue Zou[1,2][†], Yingzhuo Liu[1,2], Sharrel Lee[1,2,3], Robert B Bednarczyk[1,2], Jianting Sheng[4], Yuliang Cao[4], Stephen TC Wong[4,5,6], Dingcheng Gao[1,2,3,7]*

[1]Department of Cardiothoracic Surgery, Weill Cornell Medicine, New York, United States; [2]Sandra and Edward Meyer Cancer Center, Weill Cornell Medicine, New York, United States; [3]Neuberger Berman Lung Cancer Center, Weill Cornell Medicine, New York, United States; [4]Systems Medicine and Bioengineering Department, Houston Methodist Cancer Center, Houston Methodist Hospital, Houston, United States; [5]Department of Radiology, Houston Methodist Cancer Center, Houston Methodist Hospital, Houston, United States; [6]Department of Pathology and Laboratory Medicine, Houston Methodist Cancer Center, Houston Methodist Hospital, Houston, United States; [7]Department of Cell and Developmental Biology, Weill Cornell Medicine, New York, United States

*For correspondence:
yi.ban@nyulangone.org (YB);
dig2009@med.cornell.edu (DG)

[†]These authors contributed equally to this work

Competing interest: The authors declare that no competing interests exist.

**Abstract** Epithelial-to-mesenchymal transition (EMT) contributes significantly to chemotherapy resistance and remains a critical challenge in treating advanced breast cancer. The complexity of EMT, involving redundant pro-EMT signaling pathways and its paradox reversal process, mesenchymal-to-epithelial transition (MET), has hindered the development of effective treatments. In this study, we utilized a Tri-PyMT EMT lineage-tracing model in mice and single-cell RNA sequencing (scRNA-seq) to comprehensively analyze the EMT status of tumor cells. Our findings revealed elevated ribosome biogenesis (RiBi) during the transitioning phases of both EMT and MET processes. RiBi and its subsequent nascent protein synthesis mediated by ERK and mTOR signalings are essential for EMT/MET completion. Importantly, inhibiting excessive RiBi genetically or pharmacologically impaired the EMT/MET capability of tumor cells. Combining RiBi inhibition with chemotherapy drugs synergistically reduced metastatic outgrowth of epithelial and mesenchymal tumor cells under chemotherapies. Our study suggests that targeting the RiBi pathway presents a promising strategy for treating patients with advanced breast cancer.

## eLife assessment

This study presents a **valuable** finding that pathways associated with ribosome biogenesis (RiBi) are activated during transition cell states and targeting ribosome biogenesis could be a viable approach to overcome EMT-related chemoresistance in BCs. The evidence supporting the claims of the authors is quite **solid**, although inclusion of additional experimental support that blocking of EMT/MET is necessary for the synergistic effect of standard chemotherapy together with RiBi blockage would have strengthened the study. The work will be of interest to scientists working on breast cancer.

## Introduction

Tumor cells exploit the transdifferentiation program of EMT to acquire aggressive properties, including anchorage-independent survival, invasion, and stemness (*Nieto et al., 2016*). Multiple growth factors (TGFβ, EGF, Wnts, etc), signaling pathways (Smad2/3, PI3K/Akt, ERK1/2, etc), EMT transcription

**eLife digest** Although there have been considerable improvements in breast cancer treatments over the years, there are still many patients whose cancerous cells become resistant to treatments, including chemotherapy. Several different factors can contribute to resistance to chemotherapy, but one important change is the epithelial-to-mesenchymal transition (or EMT for short).

During this transition, breast cancer cells become more aggressive, and more able to metastasize and spread to other parts of the body. Cells can also go through the reverse process called the mesenchymal-to-epithelial transition (or MET for short). Together, EMT and MET help breast cancer cells become resilient to treatment. However, it was not clear if these transitions shared a mechanism or pathway that could be targeted as a way to make cancer treatments more effective.

To investigate, Ban, Zou et al. studied breast cancer cells from mice which had been labelled with fluorescent proteins that indicated whether a cell had ever transitioned between an epithelial and mesenchymal state. Various genetic experiments revealed that breast cancer cells in the EMT or MET phase made a lot more ribosomes, molecules that are vital for producing new proteins. Ban, Zhou et al. found that blocking the production of ribosomes (using drugs or genetic tools) prevented the cells from undergoing both EMT and MET.

Further experiments showed that when mice with breast cancer were treated with a standard chemotherapy treatment plus an anti-ribosome drug, this reduced the number and size of tumors that had metastasized to the lung. This suggests that blocking ribosome production makes breast cancer cells undergoing EMT and/or MET less resistant to chemotherapy.

Future studies will have to ascertain whether these findings also apply to patients with breast cancer. In particular, one of the drugs used to block ribosome production in this study is in early-phase clinical trials, so future trials may be able to assess the drug's effect in combination with chemotherapies.

factors (Snail, Twist, Zeb1/2, etc), and hundreds of downstream EMT related genes are involved in the EMT program (*Nieto et al., 2016*). Such complexity leads to a wide spectrum of EMT phenotypes coexisting at different stages of tumors (*Williams et al., 2019*; *Yang et al., 2020*). The EMT-endowed features contribute to tumor heterogeneity, metastasis, and therapy resistance, making EMT an attractive therapeutic target.

Current EMT-targeting strategies focus on blocking EMT stimuli, signaling transduction, or mesenchymal features (*Williams et al., 2019*; *Yang et al., 2020*). However, these approaches may paradoxically promote the reversed process of EMT, MET, which also contributes to malignancy development (*Gao et al., 2012*; *Pei et al., 2019*). Therefore, we proposed that instead of targeting epithelial or mesenchymal phenotype, inhibiting a biological process mediating the transitions of both EMT and MET could effectively overcome the limitations of traditional strategies (*Williams et al., 2019*; *Yang et al., 2020*).

To investigate the EMT process in metastatic tumor progression, we previously developed an EMT lineage-tracing model (Tri-PyMT) by combining MMTV-PyMT, Fsp1(S100a4)-Cre, and Rosa26-mTmG transgenic mice (*Fischer et al., 2015*). This model traces EMT via a permanent RFP-to-GFP fluorescence switch induced by mesenchymal-specific Cre expression. The absence of EMT reporting in metastatic lesions in this model has sparked a lively debate about the proper definition of EMT status in tumor cells and the biological significance of EMT in tumor progression (*Williams et al., 2019*; *Brabletz et al., 2018*). Rheenen's group also posited that the Fsp1-Cre mediated EMT lineage tracing model might not accurately capture the majority of EMT events in comparison to an E cadherin-CFP model (*Bornes et al., 2019*). Although we disagree with this assessment of the Fsp1-Cre model's fidelity in tracing EMT, we acknowledge the limitations of relying on a single EMT marker to investigate the EMT contributions in tumor metastasis. Notably, using the refined EMTracer animal models, *Li et al., 2020* discovered that N-cadherin + cells, rather than the Vimintin + cells, were predominantly enriched in lung metastases. These findings with different mesenchymal-specific markers underscore the complex nature of the EMT process; metastasis formation does not necessarily require the expression of many traditional mesenchymal markers.

The Tri-PyMT model, despite its limitations in comprehensive tracing of metastasis, provides unique opportunities to study EMT's role in tumor progression and chemoresistance. In particular, the

fluorescent marker switch of the established Tri-PyMT cell line reliably reports changes in EMT pheno-types (*Fischer et al., 2015*; *Lourenco et al., 2020*). Using scRNA-seq technology, we characterized the differential contributions of EMT tumor cells in tumor progression (*Lourenco et al., 2020*). Impor-tantly, post-EMT(GFP+), mesenchymal tumor cells consistently demonstrated robust chemoresistant features compared to their parental epithelial cells (pre-EMT RFP+) (*Fischer et al., 2015*), inspiring us to further study chemoresistance using the Tri-PyMT model.

## Results

### Double+ Tri-PyMT cells mark the EMT transitioning phase

In contrast to their persistence in the epithelial state (RFP+) in vivo (*Fischer et al., 2015*; *Lourenco et al., 2020*), RFP + Tri PyMT cells actively transition to GFP + in vitro in a growth medium containing 10% FBS (*Figure 1—figure supplement 1A*). As the fluorescent switch is irreversible, GFP + cells accumulate over generations until a balanced ratio of RFP+/GFP + is reached. This phenomenon indicates that the Tri-PyMT actively reports the ongoing EMT process in an EMT-promoting culture condition. Interestingly, we observed a subpopulation of cells, which were double positive for RFP and GFP, constituting approximately 2–5% of total cells (*Figure 1A*). We posited that these RFP+/GFP+ (Doub+) cells represent tumor cells transitioning from an epithelial to a mesenchymal state, since their fluorescent marker cassette has switched to GFP expression induced by Fsp1-Cre, while pre-existing RFP protein lingers due to its tardy degradation. Indeed, immunoblotting analyses confirmed the association of the double positive fluorescence and a hybrid EMT status. Doub+ cells expressed intermediate levels of both epithelial markers (Epcam and E-cadherin) and the mesenchymal marker (Vimentin) (*Figure 1B*). Further characterization of the Doub+ cells revealed higher percentages of S and G2/M phase cells in the Doub+ population compared to the RFP + and GFP + subpopulations (*Figure 1—figure supplement 1B*).

To further investigate the differential transcriptome in these EMT transitioning cells, we performed bulk RNA sequencing analysis using flow cytometry-sorted RFP+, Doub+, and GFP+ Tri-PyMT cells. Consistently, analysis of traditional EMT marker genes revealed that Doub+ cells expressed both epithelial and mesenchymal markers (*Figure 1—figure supplement 2A*). Differentially expressed genes in Doub+ cells were divided into four clusters, Trans_Up, Trans_Down, Epithelial, and Mesen-chymal markers (*Figure 1—figure supplement 2B*). Upregulated genes within the Trans_Up cluster particularly attracted our attention, as they may represent activated pathways specific to the transi-tioning phase of EMT. Interestingly, a gene set over-representative assay showed that the KEGG_Ribo-some pathway was significantly enriched in the Trans_Up gene list (*Figure 1—figure supplement 2C*).

Together, these results indicate that Doub+ Tri PyMT cells represent an active EMT-transitioning phase, as evidenced by well-established EMT markers; specific activations of biological processes in Doub+ cells warrant further investigation for developing effective strategies to intervene in the transition.

### Ribosome biogenesis pathway is enhanced in the EMT transitioning phase

To gain a deeper understanding of transcriptome alterations and ensure adequate representation of EMT-transitioning cells, RFP+, GFP+, and Doub+ cells were sorted simultaneously via flow cytom-etry; equal numbers of each population were remixed and subject for scRNA-seq analysis (*Figure 1—figure supplement 3A*).

The *t*-SNE plot demonstrated two major clusters (*Figure 1C*): one predominantly expressed epithelial genes, while the other displayed overall mesenchymal phenotypes. Doub+ cells were inte-grated into the two major clusters, suggesting that the overall single-cell transcriptome may not be sensitive enough to identify tumor cells at the EMT-transitioning phase. We, therefore, performed cell trajectory analysis (Monocle 2) based on all EMT-related genes (EMTome *Vasaikar et al., 2021*). Five cell states related to EMT status were identified (*Figure 1D*). All of them aligned well with a specific expression pattern of epithelial and mesenchymal marker genes based on AUC values (*Figure 1E*), or individual epithelial/mesenchymal pairs, such as *Fsp1/Epcam* and *Vim/Krt18* (*Figure 1—figure supplement 3B and C*). Furthermore, we calculated the EMT pseudotime of each cell and designated State 1 (the most epithelial state) as the root (*Figure 1F*). Cells were then classified into three main

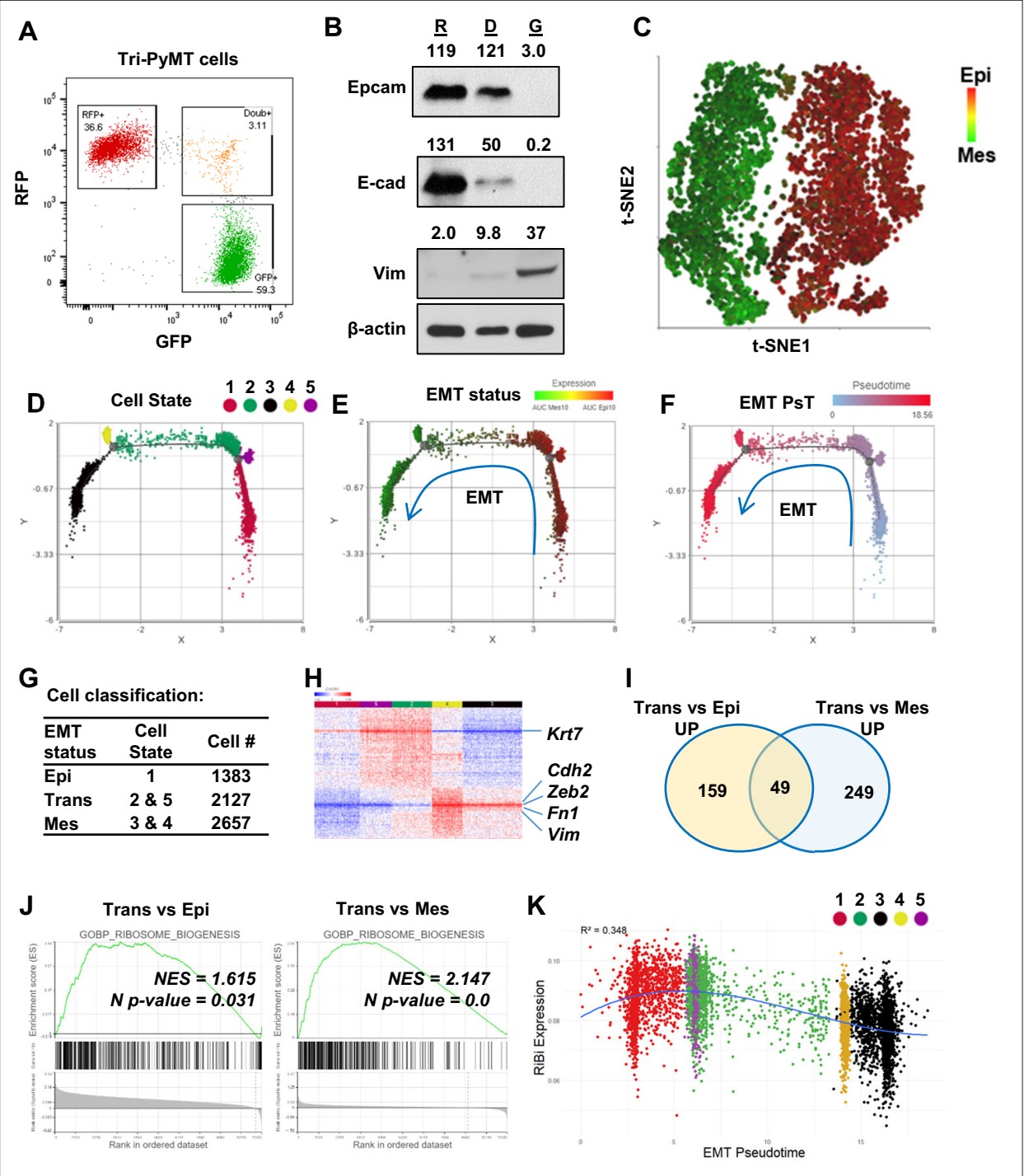

**Figure 1.** Activation of ribosome biogenesis pathway during the epithelial-to-mesenchymal transition (EMT) transitioning phase in Tri-PyMT cells. (**A**) Representative flow cytometry plot displays the percentage of RFP+, Double+ (Doub+, EMT transitioning cells), and GFP + Tri PyMT cells in culture. (**B**) Western blot of EMT markers with flow-sorted RFP+ (**R**), Doub+ (**D**), and GFP+ (**G**) Tri-PyMT cells. The Doub+ cells exhibit an intermediate EMT status with expression of epithelial marker (E-cadherin and Epcam) expression and higher mesenchymal marker (Vimentin) expression as compared with RFP + cells. The numbers indicate the normalized intensity of the band according to the β-actin band of the sample. (**C**) The t-SNE plot of scRNA-seq analysis of Tri-PyMT cells. Two major cell clusters with differential EMT status are shown with the AUC value of 10 epithelial marker genes (in red) and 10 mesenchymal marker genes (in green). The marker genes are shown in *Figure 1—source data 1*. (**D, E, F**) Trajectory analysis using Monocle DDR tree. Cell trajectory analysis was performed with filtered EMT-related genes using the Monocle 2 model. Five cell states (**D**) were identified with differential EMT statuses. From epithelial to mesenchymal phenotype, the Cell States were identified in order of 1, 5, 2, 4, and 3. EMT status (**E**) of cells was also highlighted by the AUC calculation with epithelial and mesenchymal marker genes. EMT Pseudotime (**F**) was calculated with Cell State 1 (the most

*Figure 1 continued on next page*

*Figure 1 continued*

epithelial state) as the root. (**G**) Classification of cells with EMT states. Cells were classified according to their EMT state as Epi, Trans, and Mes; the cell number in each cluster is indicated in the table. (**H**) The heatmap of differentially expressed genes in Trans cells compared to Epi and Mes cells. Totally, 313 genes were identified with criteria p<0.01, Fold change >1.2, and Average expression ≥ 5. (**I**) The common enriched GO_BP pathways with gene set enrichment analysis (GSEA) when comparing Trans *vs.* Epi and Trans *vs.* Mes. There are 49 common pathways, including five pathways related to Ribosome Biogenesis (RiBi) or rRNA processing. Pathway names are shown in *Figure 1—source data 2*. (**J**) GSEA plots showing the specific enrichment of RiBi pathway in EMT transitioning (Trans) cells compared to the Epi or Mes cells. (**K**) The scatter plot displays the correlation of RiBi activity to EMT pseudotime. The polynomial regression line (order = 3) highlights the elevated RiBi pathway in Trans phase cells, $R^2$=0.348, p<0.001.

The online version of this article includes the following source data and figure supplement(s) for figure 1:

**Source data 1.** EMT marker genes that were used in calculation of AUCell value of EMT status.

**Source data 2.** Enriched pathways in GSEAs of EMT transitioning cells.

**Source data 3.** Raw western blot images of *Figure 1B*.

**Source data 4.** Whole western blot images of *Figure 1B* with labels.

**Figure supplement 1.** Characterization of fluorescent marker switch in Tri-PyMT cells.

**Figure supplement 2.** RNA-sequencing analysis of Tri-PyMT cells.

**Figure supplement 3.** The single-cell RNA (scRNA)-sequencing analysis of Tri-PyMT cells.

**Figure supplement 4.** RT-PCR analysis of ribosome biogenesis (RiBi) genes in Tri-PyMT cells.

categories, Epi (State 1 cells), Trans (State 5&2), and Mes (State 4&3), based on their position within the EMT spectrum (*Figure 1G*). Differential gene expression analysis confirmed that Trans cells gained expression of mesenchymal markers such as *Cdh2*, *Vim*, *Fn1*, and *Zeb2*, while retaining expression of epithelial markers such as *Krt7* (*Figure 1H*).

With the differential expression gene list, we analyzed the pathway activation in Trans cells by Gene Set Enrichment Analysis (GSEA) with the biology process (BP) gene sets of the GO term (*Liberzon et al., 2015*). Among the 49 overlapped gene sets that represented significantly upregulated pathways when comparing Trans *vs.* Epi and Trans *vs.* Mes (p<0.05), six pathways were related to the ribosome or rRNA processing (*Figure 1I*, *Figure 1—source data 2*). These findings were in line with the previous bulk RNA sequencing analysis that indicated activation of RiBi pathway in EMT-transitioning (Doub+) cells. Consistently, GSEA using scRNAseq confirmed significant enrichment of RiBi genes in Trans cells compared to cells at Epi or Mes phase (*Figure 1J*). We further mapped the EMT spectrum of cells based on their EMT pseudotimes or cell states, and correlated them with their RiBi activities. A significant trend of RiBi activation was observed in the transitioning phase of EMT (*Figure 1K*, *Figure 1—figure supplement 3D*). It is worth noting that the RiBi activity is lowest in cells characterized with the latest EMT pseudotime, indicating that the elevation of RiBi activities is transient during EMT (*Figure 1K*). RT-PCR analysis also confirmed the higher expression of RiBi-related genes in Doub+ cells compared to RFP + and GFP + cells (*Figure 1—figure supplement 4*).

## Ribosome biogenesis pathway was upregulated during the MET process

The transient elevation of RiBi during EMT prompted us to ask whether the MET process required the same. Since the fluorescence switch of Tri-PyMT cells is permanent, we tracked MET by monitoring the regain of epithelial marker by post-EMT (GFP+/Epcam-) cells. We sorted GFP+/Epcam- Tri-PyMT cells via flow cytometry and injected them into *Scid* mice via tail vein. Approximately 50% of tumor cells expressed EpCam 4 weeks post-injection, indicating active MET (*Figure 2A*). We then sorted GFP+ tumor cells, including both EpCam+ and EpCam- cells, from the lungs for scRNA-seq analyses (*Figure 2—figure supplement 1A*).

Similar to our observations with in vitro cultured Tri-PyMT cells, GFP + cells from lungs formed two main clusters in the *t*-SNE plot, exhibiting overall epithelial or mesenchymal phenotypes (*Figure 2B*). We performed trajectory analysis with EMTome genes and identified nine cell states with different EMT statuses (*Figure 2C*). The relative expression of epithelial *versus* mesenchymal markers clearly illustrated the MET spectrum of tumor cells (*Figure 2D*, *Figure 2—figure supplement 1B*). By designating the most extreme mesenchymal state (State 1) as the root, we calculated the MET pseudotime of individual cells (*Figure 2E*). Consistent with the analysis for EMT, we found that tumor cells with epithelial phenotypes displayed elevated RiBi activity (*Figure 2F*, *Figure 2—figure supplement 1C*),

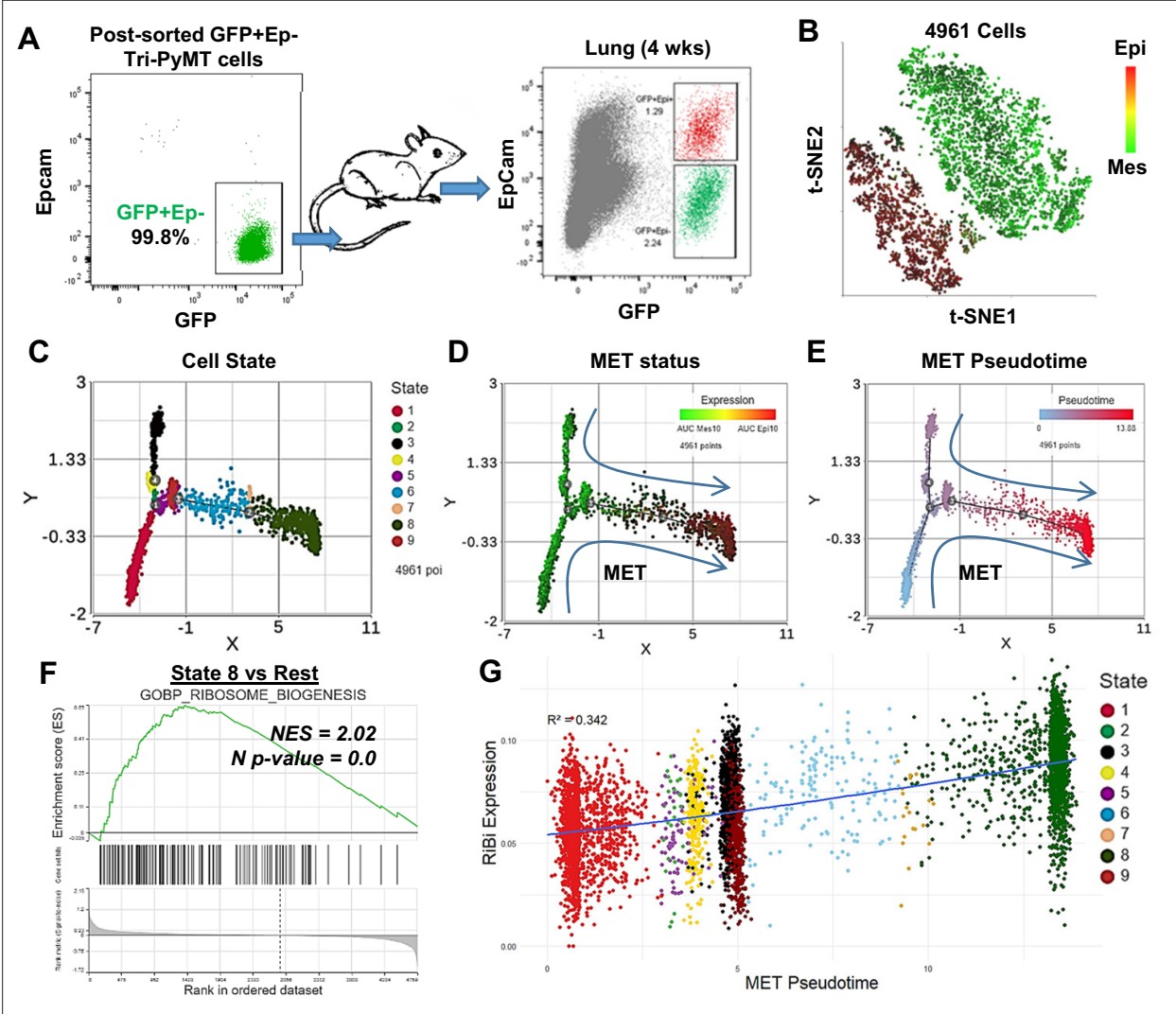

**Figure 2.** Activation of ribosome biogenesis (RiBi) pathway in mesenchymal-to-epithelial transition (MET) during lung metastasis outgrowth. (**A**) Schematic of MET induction. GFP + Tri PyMT cells were sorted by flow cytometry (left) and injected into mice through the tail vein. Metastasis-bearing lungs were harvested after 4 weeks and analyzed for the gain of epithelial marker (EpCam) in tumor cells (right). Both Epcam + and Epcam- cells were sorted and submitted for single-cell RNA sequencing (scRNA-seq) analysis. (**B**) The t-SNE plot of scRNA-seq analysis of GFP + Tri PyMT cells sorted from metastatic lungs of three individual animals. Two major cell clusters with distinct epithelial-to-mesenchymal transition (EMT) statuses were identified, as indicated by the AUC values of 10 epithelial genes (in red) and 10 mesenchymal genes (in green). (**C, D, E**) Monocle 2 model-based cell trajectory analysis identifies nine cell states (**C**). The MET process is emphasized by AUC values of mesenchymal and epithelial markers (**D**). MET Pseudotime is calculated with State 1 (the most mesenchymal state) as the root (**E**). (**F**) Gene set enrichment analysis (GSEA) plot exhibits the enrichment of the GOBP_Ribosome Biogenesis (RiBi) pathway in State 8 cells (the most epithelial phenotype) compared to the other states. (**G**) Scatter plot illustrates the correlation between RiBi activity and MET pseudotime. A polynomial regression line (order = 3) emphasizes the elevated RiBi pathway during the MET process, R2=0.342, p<0.01.

The online version of this article includes the following figure supplement(s) for figure 2:

**Figure supplement 1.** The single-cell RNA (scRNA)-sequencing analysis of GFP + Tri PyMT cells.

**Figure supplement 2.** The single-cell RNA sequencing (scRNA-seq) analysis with merged epithelial-to-mesenchymal transition (EMT) and mesenchymal-to-epithelial transition (MET) cells.

indicating its reactivation during the MET process. A significant positive correlation was detected between RiBi gene upregulation and MET pseudotime (*Figure 2G*). To eliminate the possibility that activation of the RiBi pathway was solo related to the proliferation of cells, we project the S phase score to the scatter plot of Ribi activity/MET pseudotime. Indeed, cells in the far mesenchymal state show a low S phase score, while the proliferating cells were mostly detected in the transitioning phase

and epithelial phase (*Figure 2—figure supplement 1D*). Together, these results suggested that the upregulated RiBi pathway is equally needed for mesenchymal tumor cells to undergo MET during their outgrowth in the lungs.

## Activation of ERK and mTOR signaling pathways are linked to the upregulation of ribosome biogenesis

Ribosome biogenesis was recognized as a crucial factor in cancer pathogenesis a century ago (*Pelletier et al., 2018*). To further investigate the signaling pathways responsible for RiBi upregulation in the EMT/MET process, we sorted RFP+, Doub+, and GFP+ Tri-PyMT cells and probed the signaling activations in these cells. In response to serum stimulation, Doub+ cells exhibited significantly higher levels of phosphorylated extracellular signal-regulated kinase (p-ERK) and phosphorylated mammalian target of rapamycin complex 1 (p-mTORC1) compared to either RFP+ or GFP+ cells (*Figure 3A*). The phosphorylation of Rps6, an essential ribosome protein of the 40 S subunit, is regulated by synergistic crosstalk between mTORC1 and ERK signaling (*Roux et al., 2007*; *Biever et al., 2015*). We thus explored the status of p-Rps6 and observed a significantly higher level of p-Rps6 in Doub+ cells (*Figure 3A*). These results imply a connection between differential signaling transductions and ribosome activities in EMT transitioning phase cells.

The primary function of ribosomes is to synthesize proteins to support essential biological functions (*Ruvinsky and Meyuhas, 2006*). Indeed, enhanced ribosome activity in Doub+ cells translated into distinct phenotypes in cell growth and nascent protein synthesis. Flow cytometry analysis showed that Doub+ cells possessed enlarged cell sizes compared to RFP+ or GFP+ cells (*Figure 3B*). Using the O-propargyl-puromycin (OPP) incorporation assay, we found that Doub+ cells exhibited an increased rate of nascent protein synthesis (*Figure 3C*). Another hallmark of ribosome activity is rRNA transcription, which occurs in nucleoli. By immunostaining Fibrillarin, a nucleolar marker (*Ochs et al., 1985*), we found that the Doub+ cells had significantly more nucleoli compared to cells in epithelial (RFP+) and mesenchymal (GFP+) states (*Figure 3D*). Consistently, using the EU incorporation assay, we found significantly higher transcription activity in Doub+ cells than in RFP + and GFP + cells (*Figure 3—figure supplement 1*).

Collectively, these results suggest that the elevation of the RiBi pathway in cells at the EMT transitioning phase is associated with aberrant activation of ERK and mTOR signalings, which may, in turn, confer nascent protein synthesis capability to tumor cells for completing phenotypic changes.

## Suppression of ribosome biogenesis reduced the EMT/MET capability of tumor cells

The transcription of ribosomal RNA (rRNA) is mediated by RNA polymerase I (Pol I) in eukaryotic cells (*Moss and Stefanovsky, 2002*). Small molecules such as BMH21 and CX5461 are specific Pol I inhibitors, which inhibit rRNA transcription and disrupt ribosome assembly (*Haddach et al., 2012*; *Wei et al., 2018*), providing a specific RiBi targeting strategy. We then evaluated whether these Pol I inhibitors would affect the EMT/MET process.

Fluorescence switches from RFP + to GFP + of TriPyMT cells were employed to investigate the impact of Pol I inhibitors on EMT. RFP+/Epcam + cells were sorted via flow cytometry and served as cells in an Epithelial state. In contrast to the vehicle-treated RFP+/Epcam + cells, which transitioned to GFP+ upon serum stimulation, most cells treated with either BMH21 or CX5461 stayed in the RFP+ state (*Figure 4A*, *Figure 4—figure supplement 1A*). Of note, the treatment of BMH21 or CX5461 indeed inhibited the transcription activity in cells as detected by EU incorporation assay (*Figure 4—figure supplement 1B*). To confirm that the impaired fluorescence switch of RFP+ Tri-PyMT cells is associated with the retention of epithelial phenotypes, we performed immunoblotting of EMT markers. Indeed, these Pol I inhibitors significantly blocked the expression of mesenchymal markers, including Vimentin and Snail, while preserving the expression of the epithelial marker (E-cadherin) (*Figure 4B*).

Given that the elevated RiBi activity occurs during the MET process as well, we further assessed the impact of Pol I inhibitor on the retrieval of epithelial markers by mesenchymal tumor cells. The regain of EpCam expression by the sorted GFP+/EpCam- Tri-PyMT cells in a 3D culture was measured via flow cytometry. BMH21 significantly prevented the Epcam retrieval by GFP+/Epcam- Tri-PyMT cells,

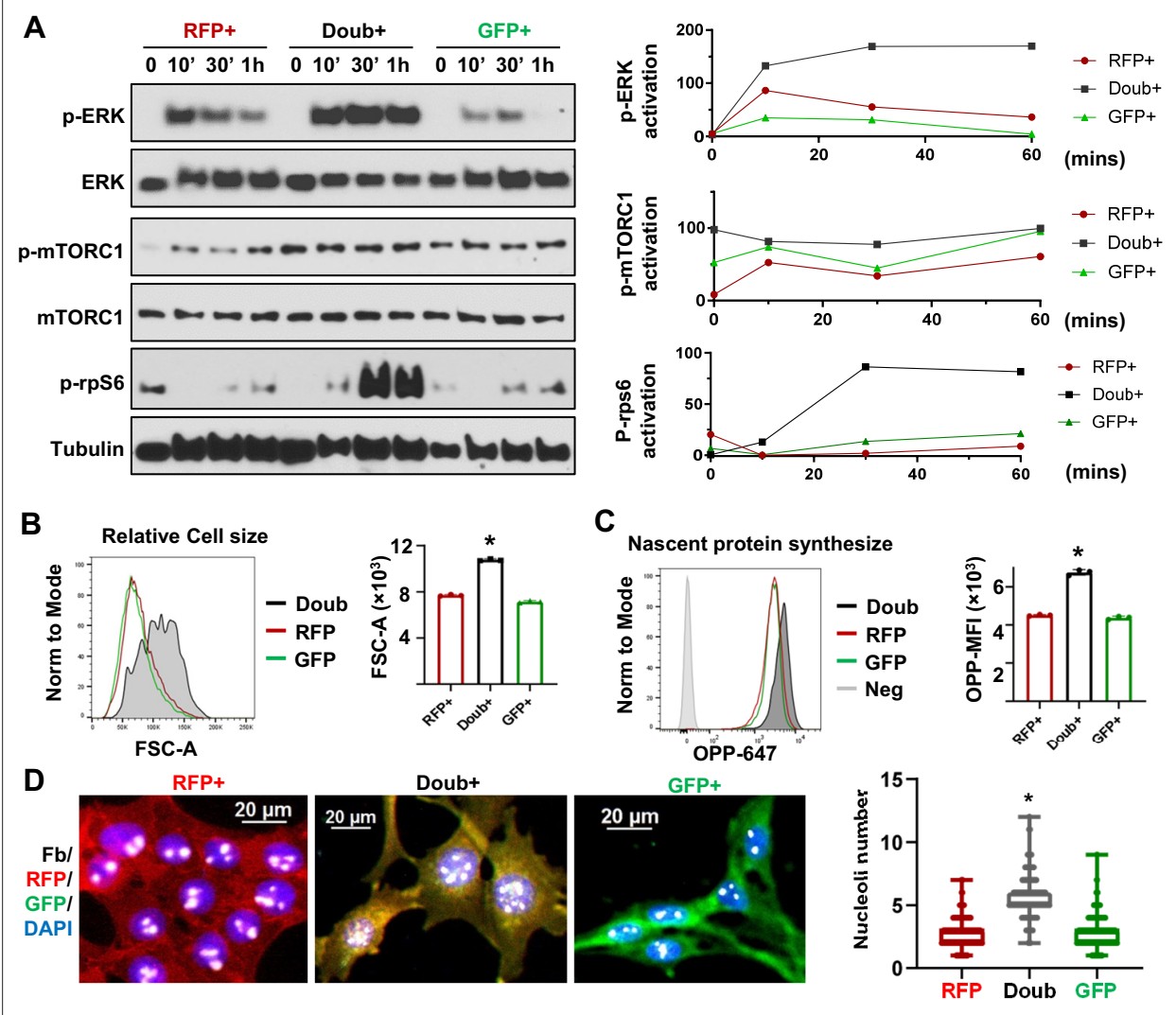

**Figure 3.** MAPK and mTOR pathways mediated ribosome biogenesis activation in epithelial-to-mesenchymal transition (EMT) transitioning phase cells. (**A**) Western blots show phospho-ERK, total ERK, phospho-mTORC1, total mTORC1, and p-rps6 in sorted RFP+, Doub+, and GFP + Tri PyMT cells following stimulation with 10% FBS at 0, 10, 30, and 60 min. Activation of p-ERK, p-mTORC1, and p-Rps6 are quantified by the band intensity after normalizing to loading controls (right). (**B**) Cell size analysis. Tri-PyMT cells (**p5**) were analyzed by flow cytometry. The relative cell size was indicated by FSC-A, for individual RFP+, GFP+, and Doub + cells. Data from three biological replicates. (**C**) Nascent protein synthesis assay. Histogram of OPP incorporation, as measured by flow cytometry, demonstrates enhanced nascent protein synthesis in Doub + cells. The rate of new protein synthesis is assessed by adding fluorescently labeled O-propargyl-puromycin (OPP). Three biological replicates, One-way ANOVA, *p<0.0001, Doub + vs. RFP+, and Doub + vs. GFP+. (**D**) Fluorescent images reveal increased ribosome biogenesis (RiBi) activity in Doub + Tri PyMT cells, as evidenced by Fibrillarin staining. RFP+, Doub+, and GFP + Tri PyMT cells were sorted and stained for nucleoli using an anti-Fibrillarin antibody. The quantification of nucleoli per cell is shown on the right. n=522 (RFP+), 141 (Doub+), 289 (GFP+). One-way ANOVA, *p<0.0001.

The online version of this article includes the following source data and figure supplement(s) for figure 3:

**Source data 1.** Raw western blot images of *Figure 3A*.

**Source data 2.** Whole western blot images of *Figure 3A* with labels.

**Figure supplement 1.** Elevated RNA transcription activity in the Doub+ Tri PyMT cells.

suggesting the impaired MET capability upon treatment (*Figure 4C*). These results laid a foundation for pharmacological inhibition of the RiBi pathway to block the EMT/MET of tumor cells.

The requirement of RiBi activity during EMT/MET transitioning was also demonstrated by genetically modulating ribosome proteins. As the organelle for protein synthesis, the ribosome comprises 4 ribosomal RNAs and approximately 80 structural ribosomal proteins (*Pelletier et al., 2018*). Depleting one r-protein usually causes decreases of other r-proteins in the same subunit, and ultimately

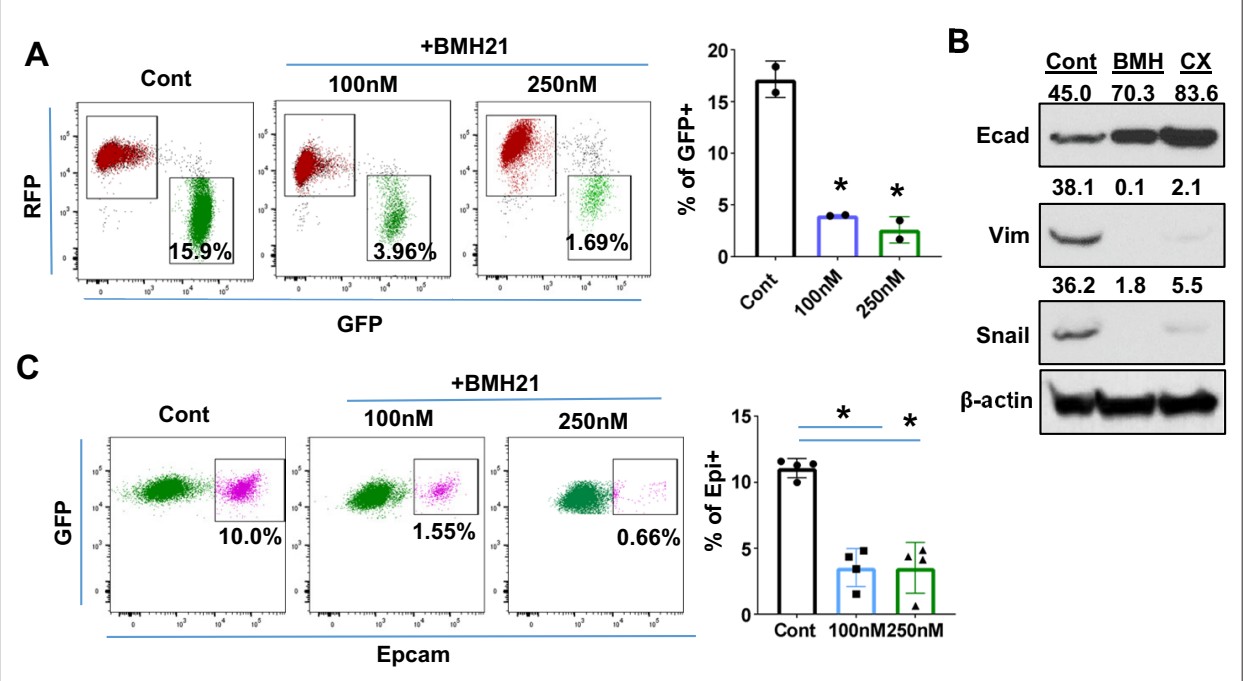

**Figure 4.** Ribosome biogenesis (RiBi) inhibition by Pol I inhibitor reduces the epithelial-to-mesenchymal transition (EMT)/mesenchymal-to-epithelial transition(MET) capability of tumor cells. (**A**) Flow cytometry plots display the percentage of GFP + cells following BMH21 treatment. RFP+/EpCam + Tri PyMT cells were sorted and cultured in a growth medium with or without BMH21 for 5 days. Percentage of GFP + cells was analyzed by flow cytometry. Biological repeat, n=2, One-way ANOVA, *p=0.0039, 100 nM *vs*. Cont; *p=0.0029, 200 nM *vs*. Cont. (**B**) Western blots reveal the expression of the epithelial marker (E-cad) and mesenchymal markers (Vim and Snail) in Tri-PyMT cells following a 5 day treatment with BMH21 (100 nM) and CX5461 (20 nM). The numbers indicate the normalized intensity of the band according to the β-actin band of the sample. (**C**) Flow cytometry plots show the percentage of Epcam+/GFP + Tri PyMT cells treated with BMH21. GFP+/EpCam- Tri-PyMT cells were sorted and cultured in 3D to induce MET for 14 days. The gain of Epcam expression was analyzed by flow cytometry. n=4, One-way ANOVA, *p<0.0001, 100 nM *vs*. Cont; *p<0.0001, 200 nM *vs*. Cont.

The online version of this article includes the following source data and figure supplement(s) for figure 4:

**Source data 1.** Raw western blot images of *Figure 4B*.

**Source data 2.** Whole western blot images of *Figure 4B* with labels.

**Figure supplement 1.** Ribosome biogenesis (RiBi)inhibition impairs RNA transcription activity and reduces the epithelial-to-mesenchymal transition (EMT) capability of tumor cells.

**Figure supplement 2.** Ribosome biogenesis (RiBi) modulating by knocking down ribosome proteins reduces the epithelial-to-mesenchymal transition (EMT)/mesenchymal-to-epithelial transition (MET) capability of tumor cells.

**Figure supplement 2—source data 1.** Raw western blot of *Figure 4—figure supplement 2A*.

**Figure supplement 2—source data 2.** Whole western blot of *Figure 4—figure supplement 2A* with labels.

compromises the overall ribosome assembly (*Robledo et al., 2008*). We, therefore, employed Lenti-shRNAs targeting Rps24 and Rps28, two essential genes of the 40 S subunit, to genetically modulate the RiBi pathway in Tri-PyMT cells (*Figure 4—figure supplement 2A*). Effective knocking-down of Rps24 or Rps28 significantly reduced the number of nucleoli (*Figure 4—figure supplement 2B*). Accompanying the downregulated RiBi activities were the impeded EMT (RFP to GFP switch) and the similarly reduced MET (regain of Epcam) (*Figure 4—figure supplement 2C and D*). These results suggested that the elevated RiBi activities are critical for tumor cells to maintain their abilities to shift between epithelial and mesenchymal states.

## RiBi inhibition synergizes with chemotherapy drugs

To assess the overall impact of Pol I inhibitor on both epithelial and mesenchymal tumor cells, we treated unsorted Tri-PyMT cells (containing both RFP + and GFP + populations) with BMH21 for 7 days. Less accumulation of GFP + cells was observed with BMH21 treatment compared with untreated

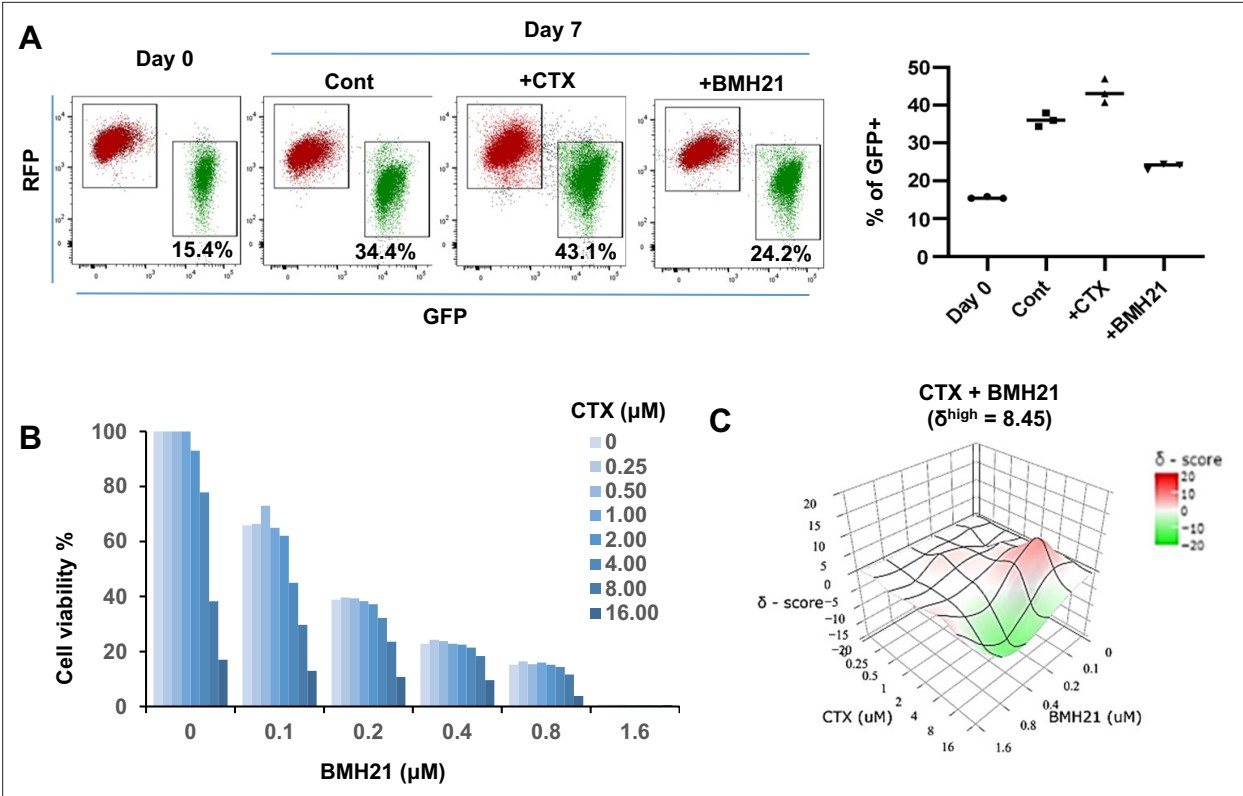

**Figure 5.** RNA Pol I inhibitor synergizes with chemo drug in vitro. (**A**) Flow cytometry plots display the fluorescence switch of Tri-PyMT cells. Tri-PyMT cells (**p5**) were treated with cyclophosphamide (CTX, 2 μM), Pol I inhibitor (BMH21, 0.1 μM), or vehicle control for 7 days. Flow cytometry was performed to analyze the percentage of GFP + cells. n=3 wells/treatment, One way ANOVA, *p=0.0050, Cont *vs.* CTX; *p=0.0002, Cont *vs.* BMH21. (**B**) Cytotoxic assay of BMH21 and CTX treatments. Tri-PyMT cells were treated with serial concentrations of BMH21 (0, 0.1, 0.2, 0.4, 0.8, and 1.6 μM) in combination with serial concentrations of CTX (0, 0.25, 0.5, 1.0, 2.0, 4.0, 8.0, 16.0 μM). Bars represent the mean value of cell viabilities from duplicated treatments, n=2 wells/treatment. (**C**) Synergy plots of BMH21 and Cyclophosphamide (CTX). Synergic scores (δ) for each combination were calculated using the ZIP reference model in SynergyFinder 3.0. Deviations between observed and expected responses indicate synergy (red) for positive values and antagonism (green) for negative values.

The online version of this article includes the following figure supplement(s) for figure 5:

**Figure supplement 1.** Combination therapy of RNA Pol I inhibitor and chemo drugs.

controls (*Figure 5A*). In contrast, treatment with the chemotherapy drug, cyclophosphamide (CTX), resulted in more GFP + cells (*Figure 5A*), consistent with our previous findings that chemoresistant features against CTX were acquired through EMT (*Fischer et al., 2015*).

Based on these observations, we hypothesized that the blockade of EMT by Pol I inhibitor would reduce the EMT-mediated chemoresistance. The combination therapies of Pol I inhibitor and chemo drugs were therefore tested. We treated Tri-PyMT cells, which contain approximately 15% GFP + cells, with a series of BMH21 concentrations with or without CTX. Cytotoxic assay after 3 days of treatment revealed an enhanced sensitivity of tumor cells to the combination therapies (*Figure 5B*). Interestingly, BMH21 and CTX exhibited optimal synergy (*Figure 5C*). Particularly at concentrations of 100–200 nM, BMH21 showed significantly high synergy scores with CTX ($\delta^{high}$ = 8.45). Such low concentrations of BMH21 were approximately 10–20% of the $IC_{max}$ in Tri-PyMT cells and have also been shown to effectively block the EMT/MET process, suggesting the synergy was likely induced by EMT blockade, rather than cytotoxicity of BMH21. Importantly, the synergic effect between BMH21 and chemotherapy drugs is not limited to CTX. We performed the combination treatment of BMH21 with the most commonly used chemo drugs of breast cancer therapy, including 5FU, Cisplatin, Doxorubicin, Gemcitabine, and Paclitaxol. Trends toward synergies were also found with most of them, especially with lower concentrations of BMH21(100–200 nM) (*Figure 5—figure supplement 1*). These results suggest that RiBi blockade by the Pol I inhibitor may represent an effective approach to overcome EMT-related chemoresistance.

# RiBi inhibition diminished chemoresistant metastasis of breast tumor cells in the lung

Both the reduced EMT-mediated chemoresistance and the diminished MET during metastatic outgrowth upon BMH21 treatment encouraged us to evaluate the efficacy of combination therapy for treating animals bearing metastatic breast tumors.

We established a competitive metastasis assay by injecting an equivalent number of GFP+ and RFP+ Tri-PyMT cells (GFP: RFP, 1:1, representing epithelial and mesenchymal tumor cells, respectively)

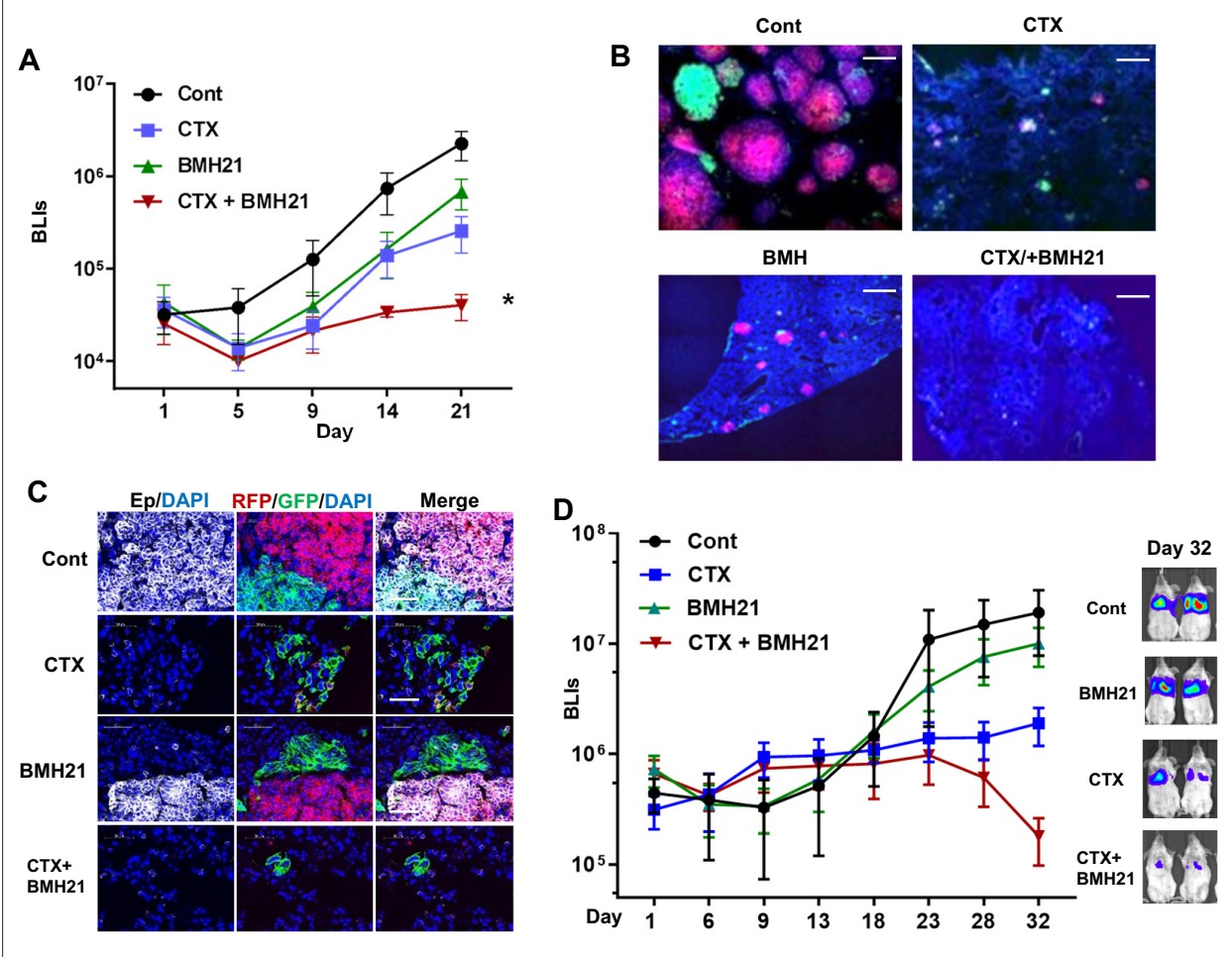

**Figure 6.** RNA Pol I inhibition synergizes with chemo drug in treating lung metastatic tumors. (**A–C**) The same number of RFP + and GFP + Tri PyMT cells were injected in animals via tail vein. Animals were treated with cyclophosphamide (CTX) (100 mg/kg, once a week, i.p.), BMH21 (20 mg/kg, five times a week, i.p.), or both in combination. (**A**) Lung metastasis growth curves as measured by bioluminescent imaging (BLI). n=5, two-way ANOVA with Tukey's test, *p=0.0154, Cont *vs.* CTX + BMH; *p=0.0151, BMH21 *vs.* CTX + BMH21; *p=0.0372, CTX *vs.* CTX + BMH21 at Day 21 post-inoculation. (**B**) Fluorescent images display RFP + and GFP + metastatic nodules in the lungs treated with CTX and BMH21. Cell nuclei were stained with DAPI. Scale bar: 200 μm. (**C**) Fluorescent images exhibit epithelial-to-mesenchymal transition (EMT) statuses of RFP + and GFP + metastases. Sections were stained with anti-EpCam antibody; cell nuclei were stained with DAPI. Scale bar: 50 μm. (**D**) Lung metastasis growth curves for LM2 model. Lung metastatic breast tumor cells (LM2 cells) were injected into animals via the tail vein. Animals were treated with CTX (100 mg/kg, once a week, i.p.), BMH21 (20 mg/kg, five times a week, i.p.), or both in combination. n=5, two-way ANOVA with Tukey's test, *p=0.0404, Cont *vs.* CTX + BMH21; *p=0.0161, BMH21 *vs.* CTX + BMH21; *p=0.0187, CTX *vs.* CTX + BMH at Day 32 post-inoculation. Representative bioluminescent imaging (BLI) images of Day 32 were presented on the right.

The online version of this article includes the following figure supplement(s) for figure 6:

**Figure supplement 1.** Differential impact of Pol I inhibitor and chemo drug on epithelial and mesenchymal Tri-PyMT cells in vivo.

**Figure supplement 2.** E-cadherin expression by LM2 cells in lung metastases.

**Figure supplement 3.** Correlation of ribosome biogenesis pathway and epithelial-to-mesenchymal transition (EMT) status in human breast cancer cells.

**Figure supplement 4.** Survival curves of breast cancer patients with different ribosome biogenesis (RiBi) activities.

into *Scid* mice via the tail vein. Tumor-bearing mice received the vehicle, single, or combination therapy of BMH21 (25 mg/kg, five times/week for 3 weeks, *ip.*) and CTX (100 mg/kg, once a week for 3 weeks, *i.p.*). Bioluminescent imaging (BLI) revealed that the combination treatment exhibited the highest restraint on the chemoresistant outgrowth of metastatic tumors compared to the vehicle, BMH21, or CTX mono-treatment groups (*Figure 6A*). To analyze the differential impacts of the therapies on epithelial and mesenchymal tumor cells, we quantified the residual RFP + and GFP + cells by flow cytometry analysis. Both RFP + and GFP + cells were significantly decreased in mice receiving the combination therapy of BMH21 and CTX (*Figure 6—figure supplement 1*). Microscopic analyses also revealed that both RFP+ and GFP+ cells grew into macrometastases in the lungs of vehicle-treated animals (*Figure 6B*). Notably, many GFP+ cells displayed epithelial markers, such as EpCam (*Figure 6C*), indicating that a MET process was involved during the outgrowth of lung metastasis. CTX treatment eliminated the majority of

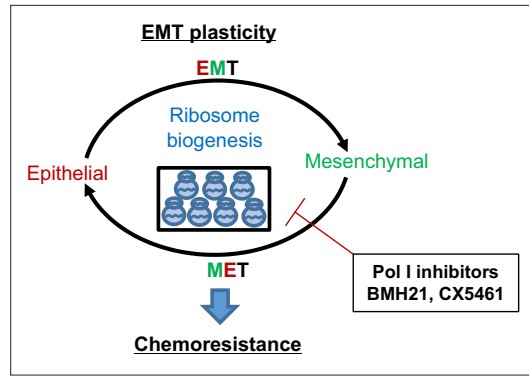

**Figure 7.** Working model of ribosome biogenesis in EMP plasticity. Tumor cells exhibit plasticity by undergoing EMT (epithelial-mesenchymal transition) and MET (mesenchymal-epithelial transition), contributing to the development of resistance to chemotherapies. Elevated ribosome biogenesis is essential for maintaining this EMT plasticity. Inhibition of ribosome biogenesis by RNA Pol I inhibitors synergizes with chemotherapeutic drugs to overcome resistance development.

RFP+/epithelial metastases, while the GFP+/mesenchymal cells showed survival advantages under chemotherapy, resulting in a higher ratio of GFP: RFP. BMH21 treatment inhibited metastatic tumor growth. Interestingly, most tumor cells under BMH21 treatment kept their EMT phenotypes as RFP+/Epcam+ or GFP+/Epcam- (*Figure 6C*), indicating the impaired EMT/MET transitioning by RiBi inhibition. Importantly, the combination of BMH21 and CTX eliminated most tumor cells including both RFP + and GFP + cells, and significantly inhibited the outgrowth of metastatic nodules under chemotherapy (*Figure 6B and C*).

We further established an experimental lung metastasis model with human breast cancer cells (MDA-MB231-LM2). The LM2 cells predominantly exhibited a mesenchymal phenotype in vitro and gained Ecad expression during the outgrowth of lung nodules in vivo (*Figure 6—figure supplement 2*), indicating the involvement of the MET process. Consistent with the Tri-PyMT model, BMH21 synergized with CTX, leading to a significantly lower metastatic LM2 tumor burden than the mock or mono-treatment groups (*Figure 6D*).

To further investigate the potential association of RiBi activity with EMT status of tumor cells in human breast cancer, we analyzed scRNA-seq data of primary tumor cells from two breast cancer patients (GSE 198745). A trend of relatively higher RiBi activity was detected in tumor cells with lower EMT pseudotime (*Figure 6—figure supplement 3*). Interestingly, a pattern showing the highest RiBi activity in EMT/MET transitioning phase was detected in the sample of patient B (*Figure 6—figure supplement 3F*), indicating the similar role of RiBi upregulation in EMT status of breast cancer patients. To examine whether RiBi activity is associated with clinical outcomes of breast cancer patients, we analyze the RNAseq data in cBioPortal databases (https://www.cbioportal.org), including the TCGA PanCancer Atlas (1084 samples) and METABRIC (2500 samples). RiBi activities of tumors were quantified by the average z-scores of genes in the RiBi pathway (303 genes). Patient samples with a score >1 were denoted as RiBi$^{High}$, while patients with a score <0.5 were denoted as RiBi$^{Low}$. The analyses of the survival data showed a significantly worse prognosis in the RiBi$^{High}$ group compared to the RiBi$^{Low}$ group (*Figure 6—figure supplement 4*).

In summary, these results suggest that inhibition of RiBi activities diminished EMT/MET transitioning capability of tumor cells. In combination with chemotherapy drugs, RiBi inhibition significantly reduced the outgrowth of chemoresistant metastasis. These results suggested that targeting the RiBi-mediated EMT/MET process may provide a more effective therapeutic strategy for advanced breast cancer.

## Discussion

Targeting EMT for cancer therapy has been challenging due to the complexity of the EMT process and controversies of promoting MET which also favors tumor progression. Using the EMT-lineage-tracing model, we found an upregulation of the RiBi pathway at the transitioning phase of both EMT and MET (*Figure 7*). The transient activation of the RiBi pathway during the EMT process has been reported previously. Prakash et al. found that elevated rRNA synthesis/RiBi pathway was concomitant with cell cycle arrest induced by TGFβ, fueling the EMT program in breast tumor cells (*Prakash et al., 2019*). Using the Tri-PyMT model, we found that the EMT transitioning (Doub+) cells had higher percentages of cells in the S and G2/M phases compared to the RFP + and GFP + cells. This is inconsistent with the observation that RiBi activity was higher in G1-arrested cells treated with TGFβ (*Prakash et al., 2019*). This discrepancy may be due to the different EMT stimuli used in the experiment systems. Additionally, further investigations are needed to determine whether a cell could complete the EMT process within a single cell cycle or requires multiple cell divisions.

A more recent study identified a subpopulation of circulating tumor cells (CTCs) in which high RiBi activities persisted to maintain their high metastatic potentials (*Ebright et al., 2020*). Interestingly, the RiBi activity was associated with epithelial phenotypes rather than mesenchymal ones in CTCs (*Ebright et al., 2020*). Indeed, EMT induction by TGFβ primarily suppressed ribosome gene expression and global translational activity (*Ebright et al., 2020*). By employing our unique EMT-lineage-tracing model, we discovered that the RiBi pathway was transiently elevated during the transitioning phases of EMT/MET program. The enhanced activation of the RiBi pathway diminished as tumor cells accomplished phenotype changes. In general, a lower RiBi activity was observed in the mesenchymal tumor cells than in the epithelial ones. Importantly, the involvement of unwonted RiBi activities during both EMT and MET processes makes RiBi a new and better target for overcoming EMT-related chemoresistance and chemoresistant metastasis.

Targeting the RiBi pathway by RNase Pol I inhibitor impaired the EMT/MET transitioning capability of tumor cells and significantly synergized with common chemotherapeutics. These observations also suggest that malignant cells may require a certain ease to 'ping pong' between epithelial and mesenchymal states to adapt to the challenging microenvironment. Of note, some commonly used chemotherapeutics (Cisplatin, 5FU, Doxorubicin, etc), although primarily targeting DNA duplications, may also affect the RiBi pathway by inhibiting rRNA processing (*Burger et al., 2010*). Therefore, the synergic effects of Pol I inhibitor varied among different combinations. Moreover, RiBi is a process that is dysregulated in most, if not all, cancers. Its involvement in the EMT/MET process makes it a feasible targeted pathway for treating patients with advanced breast cancer.

## Materials and methods

### Cell lines and cell culture

Tri-PyMT cells were established in our lab from primary tumors of MMTV-PyMT/Fsp1-Cre/Rosa26-mTmG transgenic mice (*Fischer et al., 2015*). For experiments, we used Tri-PyMT cells from passage 5 (p5) to passage 10 (p10) which contains both RFP +and GFP + cells. MDA-MB231-LM2 cells were a gift from Dr. Joan Massague. To facilitate in vivo imaging, both cell lines were genetically labeled with luciferase by lenti-Puro-Luc. Cells were cultured in DMEM with 10% FBS, 1% L-Glutamine, and 1% Penicillin Streptomycin. A routine assay for Mycoplasma (Universal Mycoplasma Detection Kit, ATCC, Cat#30–1012 K) was performed to avoid contaminations.

### Animals and tumor models

All animal works were performed following IACUC-approved protocols at Weill Cornell Medicine. CB-17 SCID mice were obtained from Charles River (Wilmington, MA). For the experimental metastasis model, RFP + and GFP + Tri PyMT cells were sorted from the 5-10[th] passage cells and re-mixed at a ratio of 1:1. Total cells ($1.5 \times 10^5$ cells) were injected through the tail vein in 100 µL of PBS in 10-week-old females. For animals subjected to chemotherapy, Cyclophosphamide (CTX, 100 mg/kg, Sigma-Aldrich, Cat# C0768) was administered once per week, i.p., for 3–4 weeks. BMH21 (25 mg/kg, Sellechehem, Cat#S7718) was administrated five times/week, i.p., for 3–4 weeks. The progression of lung metastasis tumors was monitored by bioluminescent imaging (BLI) every 3–5 days on the Xenogen IVIS system coupled with analysis software (Living Image; Xenogen).

## Flow cytometry and cell sorting

For cultured cells, single-cell suspensions were prepared by trypsinization and neutralizing with the growth medium containing 10% FBS. For the metastatic lungs, cell suspensions were prepared by digesting tissues with an enzyme cocktail containing collagenase IV (1 mg/mL), hyaluronidase (50 units/mL), and DNase I (0.1 mg/mL) in Hank's Balanced Salt Solution containing calcium (HBSS, Gibco) at 37 °C for 20–30 min. Cells were filtered through a 40 µm cell strainer (BD Biosciences) and stained with anti-Epcam antibody (G8.8, Biolegend), if needed, following a standard immunostaining protocol. SYTOX Blue (Invitrogen) was added to the staining tube in the last 5 min to facilitate the elimination of dead cells.

Samples were analyzed using the BD LSRFortessa Flow Cytometer coupled with FlowJo_v10 software (FlowJo, LLC). GFP+ and RFP+ cells were detected by their endogenous fluorescence. Flow cytometry analysis was performed using a variety of controls including isotype antibodies, and unstained and single-color stained samples for determining appropriate gates, voltages, and compensations required in multivariate flow cytometry.

For fluorescence-activated cell sorting for in vitro culture or tail vein injection, we used the Aria II cell sorter coupled with FACS Diva software (BD Biosciences). Cell preparation was performed throughout sorting procedures under sterile conditions. The purity of subpopulations after sorting was confirmed by analyzing post-sort samples in the sorter again.

## RNA-sequencing analysis

Total RNA was extracted from sorted RFP+, GFP+, and Double + Tri PyMT cells with the RNeasy Plus Kit (Qiagen). RNA-Seq libraries were constructed and sequenced following standard protocols (Illumina) at the Genomics and Epigenetics Core Facility of WCM. RNA-seq data were analyzed with customized Partek Flow software (Partek Inc). In brief, the RNA-seq data were aligned to the mouse transcriptome reference (mm10) by STAR after pre-alignment QA/QC control. Quantification of gene expression was performed with the annotation model (PartekE/M) and normalized to counts per million (CPM). Differential gene expression was performed with the Gene Specific Analysis (GSA) algorithm, which applied multiple statistical models to each gene to account for its varying response to different experimental factors and different data distribution.

For heatmap visualizations, Z scores were calculated based on normalized per-gene counts. Algorisms for the biological interpretation of differential expressions between samples such as GSEA and Over-representation assay, are also integrated into the Partek Flow platform. Gene sets of interests were downloaded from the Molecular Signatures Database (MSigDB, https://www.gsea-msigdb.org/gsea/msigdb). The raw and processed bulk RNA-seq data were available at Gene Expression Omnibus (GSE178576).

## Single-cell RNA-sequencing analysis

Single-cell suspensions were prepared following protocols from 10 X Genomics. RFP+, GFP+, and Double+ Tri-PyMT cells were sorted by flow cytometry, and cells with >90% viability were submitted for sc-RNAseq at the Genomics and Epigenetics Core Facility at WCM. Single-cell libraries were generated using 10 X Genomics Chromium Single-cell 3' Library RNA-Seq Assays protocols targeting 8,000 cells from each fraction were sequenced on the NovaSeq sequencer (Illumina). The scRNA-seq data were analyzed with the Partek Flow software (Partek Inc), an integrated user-friendly platform for NGS analysis based on the Seurat R package (*Butler et al., 2018*). The raw sequencing data were aligned to the modified mouse transcriptome reference (mm10) containing RFP, GFP, PyMT, and Cre genes by STAR. The deduplication of UMIs, filtering of noise signals, and quantification of cell barcodes were performed to generate the single-cell count data. Single-cell QA/QC was then controlled by the total counts of UMIs, detected features, and the percentage of mitochondria genes according to each sample (*Figure 1—figure supplement 3A*, *Figure 2—figure supplement 1A*). The top 20 principal components (PC) were used for tSNE visualization. Differential gene expression analysis, GSEA, Geneset Overrepresentation assay, and Cell Trajectory analysis (Monocle 2 model) are also integrated into the Partek Flow platform (Partek Inc).

To highlight the overall expression of feature genes in a pathway, such as ribosome biogenesis or EMT status, we calculated the AUCell values for the gene list. AUCell calculates a value for each cell by ranking all genes based on their expression in the cell and identifying the proportion of the gene

list that falls within the top 5% of all genes. For the epithelial or mesenchymal gene lists, we selected genes based on their overall expression levels in Tri-PyMT cells, ensuring consistency with their reported associations to epithelial or mesenchymal phenotypes in the literature (*Figure 1—source data 1*). For RiBi genes, we used the ribosome biogenesis pathway gene list from the GOBP_Ribosome_Biogenesis (GO:0042254) in MSigDB.

The raw and processed scRNA-seq data were available at Gene Expression Omnibus (GSE178577 and GSE178578).

## Tissue processing, Immunofluorescence, and Microscopy

Lungs with metastases were fixed in 4% paraformaldehyde overnight, followed by immersion in 30% sucrose for two days. They were then embedded in the Tissue-Tek O.C.T. compound (Electron Microscopy Sciences). Serial sections (10–20 μm, at least 10 sections) were prepared for immunofluorescent staining.

For staining cultured or sorted cells, $2 \times 10^3$ cells/well were seeded in eight-well chamber slides (Nunc Lab-Tek, Thermo Fisher) and cultured overnight in a growth medium. Standard immunostaining protocols were followed using fluorescent-conjugated primary antibodies. Fluorescent images were captured using a Zeiss fluorescent microscope (Axio Observer) with Zen 3.0 software (Carl Zeiss Inc).

## Western blot analysis

Cells were homogenized in 1 x RIPA lysis buffer (Millipore) containing protease and phosphatase inhibitors (Roche Applied Science). The samples were then boiled in 1 x Laemmli buffer with 10% β-mercaptoethanol and loaded onto 12% gradient Tris-Glycine gels (Bio-Rad). Western blotting was performed using the antibodies listed in the antibody table. Quantification of the Western blots was carried out using ImageJ. The relative intensity of each band was normalized to that of β-actin or tubulin, serving as loading controls for the same blot.

## Antibodies used in the experiments

| Species | Antigen | Cat#, Company | Dilution | Application |
|---|---|---|---|---|
| Mouse | E-Cadherin | 144725, Cell Signaling | 1:40 | WB |
| Mouse | EpCAM | G8.8 Biolegend | 1:100 | FC, WB, IF |
| Mouse | Vim | 5741T, Cell Signaling | 1:500 | WB |
| Mouse | β-actin | Ab8227, Abcam | 1:2000 | WB |
| Mouse | Fibrillalin | Ab5821, Abcam | 1:100 | IF |
| Mouse | p-ERK | 4370, Cell Signaling | 1:1000 | WB |
| Mouse | ERK | 4695, Cell Signaling | 1:1000 | WB |
| Mouse | p-mTORC1 | 3895 s, Cell Signaling | 1:1000 | WB |
| Mouse | mTORC1 | 2983T, Cell Signaling | 1:1000 | WB |
| Mouse | p-rps6 | 4858T, Cell Signaling | 1:1000 | WB |
| Mouse | Tublin | 11224–1-AP, Proteintech | 1:1000 | WB |
| Mouse | Snail | 3895, Cell Signaling | 1:500 | WB |

## Cell viability assays

To determine the viability of Tri-PyMT cells under chemotherapy, cells ($2 \times 10^3$ cells/well) were seeded in 96-well adherent black-walled plates, and treated with a serial concentration of 5-FU, Cisplatin, 4-Hydroperoxy Cyclophosphamide, Doxorubicin, Gemcitabine, and Paclitaxel together with BMH21, for 72 hr. After treatment, cell viability was measured using the CellTiter-Glo Luminescent Cell Viability Assay (Promega).

To quantify the cytotoxic effect and potential synergic effects of drug combinations, we used SynergyFinder 3.0 (*Ianevski et al., 2020*) for data analysis. Basically, normalized cell viability data

were formatted and analyzed with the online server at SunergyFinder (https://synergyfinder.fimm.fi). The LL4 and ZIP methods were chosen for curve fitting and synergy calculation, respectively.

## Cell proliferation, transcription activity, and nascent protein translation assays

To characterize cellular activities in nascent DNA, RNA, and protein synthesis, we applied EdU (5-ethynyl-2'-deoxyuridine), EU (5-ethynyl uridine), and OPP (O-propargyl-puromycin) incorporation assays, respectively. Cells ($3\times10^5$ cells/well) were seeded in a six-well plate and treated (or left untreated) according to experimental settings. Cells were labeled with EdU (10 µM, for 30 min), EU (1 mM, for 1 hr), or OPP (10 µM, for 30 min) in the incubator. After labeling, cells were harvested by trypsinization, fixed with 3.7% formaldehyde in PBS, and permeabilized with 0.5% Triton-X100 in PBS. Labeling was detected using the Click-&-Go detection kit (Vector Laboratories) following the standardized protocol and analyzed by flow cytometry.

## RT-PCR analysis

Total RNA was extracted from sorted RFP+, GFP+, and Doub+ Tri PyMT cells using the RNeasy Plus Kit (Qiagen). For cDNA synthesis, 100 ng of RNA was used with the qScript cDNA SuperMix (Quanta Bio). Q-PCR was performed using SsoAdvanced Universal SYBR Green Supermix (Bio-Rad) with target gene-specific primers (as shown in the table) and Gapdh as the housekeeping control. The PCR protocol included an initial denaturation at 98 °C for 2 min, followed by 40 cycles of 98 °C for 15 s, 60 °C for 30 s, and 72 °C for 30 s, with signal readings at the end of each cycle. This was followed by a final extension at 72 °C for 5 min and melt curve analysis on the CFX96 Real-Time System (Bio-Rad).

## Gene specific primers used in the experiments

| Target ene | Forward | Reverse |
| --- | --- | --- |
| Bop1 | GGTCTCGGAGGAAGAGCACC | ACCGCCAAATAGTCCCCTCG |
| Gapdh | GGTCCTCAGTGTAGCCCAAG | AATGTGTCCGTCGTGGATCT |
| Gemin4 | CCTCACAGGTCCACGAAGGG | TGCCCACATCCATCACCAGA |
| Its1 | TCCATCTGTTCTCCTCTCTCT | ATCGGTATTTCGGGTGTGAG |
| Its2 | CTGCCTCACCAGTCTTTCTC | ACCTCGACCAGAGCAGAT |
| Ecad | ACACCGATGGTGAGGGTACACAGG | ACACCGATGGTGAGGGTACACAGG |
| Ncad | AAAGAGCGCCAAGCCAAGCAGC | TGCGGATCGGACTGGGTACTGTG |
| Nop58 | ACAGCAGAAGCATTAGCAGCA | CGACAGCCAGAGGTTCATGG |
| Npm1 | GCGAGATCTCCTGCGACCAT | ACTTCGGTGTGGGAGAAGCC |
| Occl | TGCTAAGGCAGTTTTGGCTAAGTCT | AAAAACAGTGGTGGGGAACGTG |
| Polr1a | GGCTCTGCGCTACAAGACTC | AAGGAAATGCCCTGAAGCCG |
| Rpl29 | GCAGTGAGGGAAGCTTTTCCG | CATGTCTGCACGGTAACCCG |
| Rpl8 | ACAGAGCTGTTCATCGCAGC | ACGATCGTACCCTCAGGCAT |
| Rps24 | ACACAGTAACCATCCGGACCA | TTTTGGCCAGCTTTTCCCGA |
| Rps28 | GATATCCAGAACCCCACCAGC | AATGTCAAAGGCCCCGTTCG |
| Rps9 | GTCTCGGGCCTGAGTTCGTA | CCTGCGCAGTAAAGTGTCGT |
| S100a4 | CCTGTCCTGCATTGCCATGAT | CCCACTGGCAAACTACACCC |
| Setd4 | GGAACTGCGCGTCCTTGTG | GTAACAAAACGCCCTCGGCA |
| Snail | ACTGGTGAGAAGCCATTCTCCT | CTGGCACTGGTATCTCTTCACA |
| Utp6 | AGGGCATTTGGGGAGTAAGGG | TGCCTGGGGTCTGTCTCAGT |
| Vim | TGACCTCTCTGAGGCTGCCAACC | TTCCATCTCACGCATCTGGCGCTC |

*Continued on next page*

*Continued*

| Target ene | Forward | Reverse |
|---|---|---|
| Xpo1 | GGGAGCTTCAGCATCAGCAA | TTCCTTCCTTGTCCGGGCTT |
| Zeb1 | GATTCCCCAAGTGGCATATACA | TGGAGACTCCTTCTGAGCTAGTG |
| Zeb2 | tggatcagatgagcttcctacc | agcaagtctccctgaaatcctt |

## Statistical analysis

To determine the sample size for animal experiments, we performed power analysis assuming the (difference in means)/(standard deviation) is >2.5. Consequently, all animal experiments were conducted with ≥5 mice per group to ensure adequate power for two-sample t-test comparison or ANOVA. Animals were randomized within each experimental group, and no blinding was applied during the experiments. Results were expressed as mean ± SEM. Data distribution within groups and significance between different treatment groups were analyzed using GraphPad Prism software. p-values <0.05 were considered significant. Error bars represent SEM, unless otherwise indicated.

The manuscript was prepared following the general recommendations of ICMJE.

## Acknowledgements

We thank Dr. Jenny Xiang of the Genomics Resources Core Facility and Jason McCormick of the Flow Cytometry Core Facility for their professional advice. We thank Dr. Vivek Mittal and Dr. Nasser K Altorki for their comments on this work. We thank Dr. Divya Ramchandani for supporting the experimental technique. This work was supported by National Cancer Institute (NCI) Grant NIH R01 CA205418 (DG) and R01 CA244413 (DG). This work was also supported by The Neuberger Berman Foundation Lung Cancer Research Center, a generous gift from Jay and Vicky Furman; and generous funds donated by patients in the Division of Thoracic Surgery to Dr. Altorki. The funding organizations played no role in experimental design, data analysis, or manuscript preparation.

## Additional information

### Funding

| Funder | Grant reference number | Author |
|---|---|---|
| National Cancer Institute | R01CA244413 | Dingcheng Gao |
| National Cancer Institute | R01 CA205418 | Dingcheng Gao |

The funders had no role in study design, data collection and interpretation, or the decision to submit the work for publication.

### Author contributions

Yi Ban, Conceptualization, Data curation, Formal analysis, Writing – original draft, Methodology; Yue Zou, Data curation, Conceptualization; Yingzhuo Liu, Data curation, Conceptualization, Methodology; Sharrel Lee, Methodology; Robert B Bednarczyk, Data curation, Methodology; Jianting Sheng, Yuliang Cao, Software, Methodology; Stephen TC Wong, Software, Validation, Methodology; Dingcheng Gao, Conceptualization, Formal analysis, Funding acquisition, Supervision, Validation, Writing – original draft, Methodology

### Author ORCIDs

Yi Ban http://orcid.org/0000-0002-7644-9686
Dingcheng Gao https://orcid.org/0000-0002-3903-2603

### Ethics

All animal works were performed in accordance with IACUC approved protocol (#2022-0030) at Weill Cornell Medicine. The protocol was approved by the Committee on the Ethics of Animal Experiments

of the Weill Cornell Medicine. All animal procedures were performed under guidelines to minimize suffering.

Reviewer #1 (Public review): https://doi.org/10.7554/eLife.89486.3.sa1
Author response https://doi.org/10.7554/eLife.89486.3.sa2

# Additional files

## Supplementary files
• MDAR checklist

## Data availability
Sequencing data have been deposited in GEO under accession codes GSE178576, GSE178577, GSE178578 and GSE178579.

The following datasets were generated:

| Author(s) | Year | Dataset title | Dataset URL | Database and Identifier |
|---|---|---|---|---|
| Gao D | 2022 | Targeting epithelial-mesenchymal plasticity via ribosome inhibition to reduce chemoresistant metastasis of breast cancer | https://www.ncbi.nlm.nih.gov/geo/query/acc.cgi?acc=GSE178579 | NCBI Gene Expression Omnibus, GSE178579 |
| Gao D | 2022 | Targeting epithelial-mesenchymal plasticity via ribosome inhibition to reduce chemoresistant metastasis of breast cancer | https://www.ncbi.nlm.nih.gov/geo/query/acc.cgi?acc=GSE178578 | NCBI Gene Expression Omnibus, GSE178578 |
| Gao D | 2022 | Targeting epithelial-mesenchymal plasticity via ribosome inhibition to reduce chemoresistant metastasis of breast cancer | https://www.ncbi.nlm.nih.gov/geo/query/acc.cgi?acc=GSE178577 | NCBI Gene Expression Omnibus, GSE178577 |
| Gao D | 2022 | Targeting epithelial-mesenchymal plasticity via ribosome inhibition to reduce chemoresistant metastasis of breast cancer | https://www.ncbi.nlm.nih.gov/geo/query/acc.cgi?acc=GSE178576 | NCBI Gene Expression Omnibus, GSE178576 |

The following previously published dataset was used:

| Author(s) | Year | Dataset title | Dataset URL | Database and Identifier |
|---|---|---|---|---|
| Liu S | 2022 | Single-cell and spatially resolved analysis uncovers cell heterogeneity of breast cancer | https://www.ncbi.nlm.nih.gov/geo/query/acc.cgi?acc=GSE198745 | NCBI Gene Expression Omnibus, GSE198745 |

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
