## [Editor Report · eLife assessment]

This study presents a **valuable** finding that pathways associated with ribosome biogenesis (RiBi) are activated during transition cell states and targeting ribosome biogenesis could be a viable approach to overcome EMT-related chemoresistance in BCs. The evidence supporting the claims of the authors is quite **solid**, although inclusion of additional experimental support that blocking of EMT/MET is necessary for the synergistic effect of standard chemotherapy together with RiBi blockage would have strengthened the study. The work will be of interest to scientists working on breast cancer.

---

## [Referee Report · Reviewer #1 (Public review)]

The process of EMT is a major contributor of metastasis and chemoresistance in breast cancer. By using a modified PyMT model that allows identification of cells undergoing EMT and their decedents via S100A4-Cre mediated recombination of the mTmG allele, Ban et al. tackle a very important question of how tumor metastasis and therapy resistance by EMT can be blocked. They identified that pathways associated with ribosome biogenesis (RiBi) are activated during transition cell states. This finding represents a promising therapeutic target to block any transition from E to M (activated during cell dissemination and invasion) as well as from M to E (activated during metastatic colonization). Inhibition of RiBi-blocked EMT also reduced the establishment of chemoresistance that is associated with an EMT phenotype. Hence, RiBi blockage together with standard chemotherapy showed synergistic effects, resulting in impaired colonization/metastatic outgrowth in an animal model. The study is of great interest and of high clinical relevance as the authors show that blocking the transition from E to M or vice versa targets both aspects of metastasis, dissemination form the primary tumor and colonization in distant organs.

The study is done with high skill using state of the art technology and the conclusions are convincing and solid, but some aspects require some additional experimental support and clarification. It remains elusive whether blocking of EMT/MET is necessary for the synergistic effect of standard chemotherapy together with RiBi blockage or whether a general growth disadvantage of RiBi treated cells independent of blocking transition is responsible. How can specific effect on state transition by RiBI block be seperated from global effects attributed to overall reduced protein biosynthesis, proliferation etc.? Some other aspects are misleading or need extension:

In the revised version, the authors appropriately addressed all my comments. I'd like to congratulate the authors for this wonderful work!

---

## [Author Response]

The following is the authors’ response to the original reviews.

**Reviewer #1 (Public Review):**
The process of EMT is a major contributor to metastasis and chemoresistance in breast cancer. By using a modified PyMT model that allows the identification of cells undergoing EMT and their decedents via S100A4-Cre mediated recombination of the mTmG allele, Ban et al. tackle a very important question of how tumor metastasis and therapy resistance by EMT can be blocked. They identified that pathways associated with ribosome biogenesis (RiBi) are activated during transition cell states. This finding represents a promising therapeutic target to block any transition from E to M (activated during cell dissemination and invasion) as well as from M to E (activated during metastatic colonization). Inhibition of RiBi-blocked EMT also reduced the establishment of chemoresistance that is associated with an EMT phenotype. Hence, RiBi blockage together with standard chemotherapy showed synergistic effects, resulting in impaired colonization/metastatic outgrowth in an animal model. The study is of great interest and of high clinical relevance as the authors show that blocking the transition from E to M or vice versa targets both aspects of metastasis, dissemination from the primary tumor, and colonization in distant organs.

We appreciate the positive acknowledgment of our work.

The study is done with high skill using state-of-the-art technology and the conclusions are convincing and solid, but some aspects require some additional experimental support and clarification. It remains elusive whether blocking of EMT/MET is necessary for the synergistic effect of standard chemotherapy together with RiBi blockage or whether a general growth disadvantage of RiBi-treated cells independent of blocking transition is responsible.

We appreciate the reviewer for raising the pertinent query regarding the interrelation between EMT/MET blocking by RiBi inhibition and its synergistic effect with chemotherapy drugs. Our experimental data suggests a potential consequence of these events. Specifically, when assessing the potency of RiBi inhibitors (BMH21 and CX5410), we observed a pronounced EMT/MET blocking effect at concentrations preceding the emergence of cytotoxic effects (refer to Fig. 4 and Supplementary Fig S8). Notably, the IC50 for BMH21 was approximately 200nM, which is a concentration surpassing those that manifested the EMT/MET blocking effects. Crucially, the enhanced synergy of RiBi inhibitors with chemotherapy drugs was predominantly seen at these lower concentrations (as illustrated in Supplementary Fig S10). Therefore, the EMT/MET blocking by RiBi inhibition, rather than the cytotoxic effect, is likely instrumental for the synergy with chemotherapy drugs. The result was highlighted in Page#16.

How can specific effects on state transition by RiBI block be separated from global effects attributed to overall reduced protein biosynthesis, proliferation etc.?

We appreciate the reviewer's insightful query. We agree that RiBi activity and associated protein synthesis are fundamental processes for cell viability, making it challenging to clearly delineate the overall effects of RiBi blockage to the specific effects of EMT state transition. Our results showed an elevated RiBi activity during the EMT transitioning phases, concomitant with enhanced nascent protein synthesis, indicating a higher-than-normal requirement of new proteins for cells to switch their phenotype. This would provide us a chance to target the excessive activities of RiBi to block EMT/MET transition. Based on a similar consideration, we chose to apply shRNA instead of CRISPR technology to modulate RiBi gene expression. By comparing to scramble controls, the growth rates of the Rps knockdown cells (both RFP+ and GFP+ cells) were not significantly affected, while the EMT/MET transitioning was impaired (Supplementary Fig 9). These results may provide evidence of uncoupling the cell proliferation and EMT/MET status changes by inhibiting RiBi pathway.

Some other aspects are misleading or need extension.
**Reviewer #1 (Recommendations For The Authors):**
(1) The analysis of RiBi expression during EMT in Fig. 1K shows that transition states have high RiBi levels, whereas E and M states are low. Analyses of MET in Fig.2G indicate that M states have the lowest, transition states upregulate RiBi while E states have the highest levels of RiBi expression. This is puzzling and how can it be explained? It would be helpful to demonstrate how these two settings are related by combining results from Figs 1 and 2 in an E-Trans-M-Trans-E state graph (in a sequence of EMT/MET). Does it mean that the initial E state starts with lower RiBi and the final E state displays the highest RiBi expression? In other words, are the initial E state and the one after MET different?

Thank the reviewer for raising the concern about which EMT/MET state exhibits the highest RiBi activity. Following the reviewer's suggestions, we merged the scRNA-seq data of EMT and MET cells and performed the trajectory analysis. Similar epithelial-mesenchymal spectrums were detected from these cells (For reviewers Fig 1). Notably, the highest RiBi activity was detected in the early EMT transitioning or the late MET transitioning cells (revised For reviewers Fig 1D). Addressing the question of the reviewer, the initial E state (of EMT cells) did not show significant differences to the final E state (of MET cells) in comparisons of EMT pseudotime and RiBi activities. In addition, the analysis with merged cells also revealed:

(1) Both the EMT (In_Vitro_Mix) and MET (In_Vivo_GFP) cells were generally divided into two major clusters representing epithelial and mesenchymal phenotypes (For reviewers Fig 1A, 1B).

(2) The EMT and MET cells exhibited similar EMT spectrums (EMT/MET status, and pseudotime) in the trajectory analysis (For reviewers Fig 1C, 1D).

(3) Cells with high RiBi activity were mostly from the transitioning cell during EMT (In_Vitro_Mix) cells (For reviewers Fig 1D).

(2) It needs to be elaborated on how the experiment in Fig. 4A was exactly done. Are there cells isolated directly from the autochthonous TriPyMT tumor in contrast to steady-state cultures from Fig. 1? Does the control graph represent 0d in culture or have the cells been cultured for the same amount of time as the treated samples? How do these observed 15% GFP+ cells are related to the 15% GFP+ cells obtained at day 0 and 34% at d7 control condition in Fig. 5A?

Following the reviewer’s suggestion, we have amended the figure legend to clarify the experiment settings. In Fig. 4A, we initiated the experiment with sorted RFP+/Epcam+ cells. The control cells were cultured for the same period of time (5 days) as drug-treated cells did. We apologize for the unclear description. The percentage of GFP+ cells in this experiment is not related to the experiment in Fig 5A, where the initial cell population comprised an unsorted mix of RFP/GFP cells.

(3) Fig. 4B: Since the bulk population is loaded in the WB, does that suggest that the epithelial state is stabilized/enhanced or does it reflect only different cell ratios? So, it would be important to show the WB for RFP+ and GFP+ cells separately.

Thank the reviewer for the query regarding Fig. 4B. We apologize for the unclear explanation. The experimental setup for Fig 4B was identical to that of Fig 4A, where the sorted RFP+ cells were utilized at the start. Indeed, the observed increase in epithelial markers and decrease in mesenchymal markers in cells treated with BMH and CX suggest a higher proportion of cells maintaining the RFP+ state.

Performing WB for RFP+ and GFP+ cells separately may not address the question we asked since the experiment was initialed with pure RFP+ cells. Also, the expression of the fluorescent markers is closely aligned with the EMT status of the cells with and without drug treatment.

(4) Figs. 4-6: The authors claim that there is less EMT under treatment. If the experiment was done over 5 days (as indicated in Fig.4b legend), it is necessary to rule out that shifts in E/M ratios are attributed to the effects of treatment on proliferation/survival affecting both populations differently. How do the same cells grow under treatment when injected orthotopically/subcutaneously?

We apologized for the unclear descriptions. The effect of blocking the transitioning of EMT with RiBi inhibitors were performed with purified RFP+/EpCam+ cells. All GFP+ cells in this experiment setting were transformed from RFP+ cells. Given the fluorescence switch was well correlated with EMT status of cells, RFP and GFP were used as EMT reporters. Similarly, we used purified GFP+/EpCam- cells as the initial population to study the MET process of tumor cells.

To address the reviewer's concern regarding how RiBi inhibition may differentially affect the growth of RFP+ and GFP+ cells, we conducted a cell cycle assay using Tri-PyMT cells, which include both RFP+ and GFP+ populations. Our results demonstrated that both RFP+ and GFP+ cells exhibited a trend towards G2/M phase accumulation when treated with BMH21. It is important to note that the impact of BMH21 on the cell cycle was less pronounced than previously reported by Fu et al. (Oncol Rep, 2017). This is likely because the dose used for EMT inhibition in our study was approximately one-tenth of the dose known to inhibit cell growth (For Reviewers Fig 2). Also, no significantly differential impacts were detected between RFP+ and GFP+ cells.

We have previously characterized the proliferation rate of RFP+ and GFP+ populations (Lourenco et al 2020). RFP+ cells proliferate faster than GFP+ cells. Primary tumor cells derived from RFP+ cells also grew faster than GFP+ tumors (Lourenco et al 2020).

(5) Fig. 6B: this image is puzzling. Only in the lower two panels the outline of the lung is visualized by DAPI staining. The upper two panels look like there is no lung tissue in ctrl (no DAPI+GFP-RFP- cells) or show almost exclusively DAPI+GFP-RFP- cells that are present in a clustered assembly. Do the latter represent lymphoid cell clusters or normal lung tissue?

To improve the clarity of fluorescent images in Fig 6B, we enlarged the merge images with higher contrast (Revised Fig. 6B). The DAPI+/RFP-/GFP- region represent normal lung tissue. Nodules with either RFP or GFP signals represent tumor lesions.

(6) Text: Several typos and sentences should be revised, including p. 3 "Le et al. discovered" which should read as "Li et al. discovered", p.8 "Vimten", p.10 "Cells were then classified cells into three main categories", GSEA should be spelled out as Gene Set Enrichment Analysis (not Assay), p. 13 "cells, suggesting the impaired MET capability with upon treatment".

We apologize for the typos. All were corrected in the revised manuscript.

(7) Figures: Color gradient indicator in Fig. 1E does not reflect the colors of the cells, Fig. S5A+C are not referenced in the text, there is mislabeling of S5B,C,D in the legend, graph in Fig. 3D is placed two times and overlapping, Fig. 6C labeling needs adjustments, labeling of Fig. 6D should be similar to Fig. 6A: CTX blue and BMH21 green.

We apologize for these errors and made corrections. Color in Fig.1E represents the EMT status of tumor cells as indicated in the revised figure, red for more epithelial, and green for more mesenchymal features. Fig S5 is now Fig S6, and referred in the revised manuscript. Legend for figures were corrected. Labels of Fig 6 were adjusted.

**Reviewer #2 (Public Review):**
(1) The current manuscript by Ban et al describes that cells undergoing EMT have increased rRNA synthesis, as analyzed by RNA seq-based gene expression analysis, and that the increased rRNA synthesis provides a therapeutic opportunity to target chemoresistance. The cells utilized in this manuscript were isolated from the authors' Tri-PyMT EMT lineage tracing model published a few years ago which demonstrated that cells undergoing EMT are not the cells that are contributing to metastasis but rather to tumor chemoresistance (Fischer, Nature 2015). This in vivo model has since then been criticized for not capturing all relevant EMT events which the authors also acknowledge in the introduction. The authors therefore reason that they use this lineage tracing model to better understand the role of EMT in chemoresistance.A major problem with the current manuscript is that the authors present many of their findings as a novel without the proper acknowledgment of previously published literature in particular, Prakash et al., Nature Communications, 2019 and Dermitt, Dev Cell, 2020. In the studies by Prakash, the authors demonstrate that maintaining ongoing rRNA biogenesis is essential for the execution of the EMT program, and thus the ability of cancer cells to become migratory and invasive. Further, Prakash et al showed that blocking rRNA biogenesis with a small molecule inhibitor, CX-5461 (which is also used in the study by Ban et al) specifically inhibits breast cancer growth, invasion, EMT, and metastasis in animal models without significant toxicity to normal tissues. As such a significant revision that is necessary at this time is a rewrite of the manuscript especially the introduction and the discussion to more accurately describe and cite previously published findings and then highlight the current work by Ban et al which nicely builds on the previously published literature as it highlights the contribution of EMT to chemoresistance rather than metastasis. The suggestion for the authors is that they therefore should focus on highlighting the chemotherapy resistance angle as their Tri-PyMT EMT lineage tracing was chosen to test this angle and as such focus on both primary tumor growth and metastasis.

We appreciate the reviewer’s insightful feedback. In response, we have revised a section in the discussion to better highlight how our study builds upon and extends the work of others. We acknowledge that the link between ribosome biogenesis (RiBi) and the epithelial-mesenchymal transition (EMT) pathway was noted by prior researches (Prakash et al. 2019; Ebright et al. 2020). In the revised manuscript, we have included extra discussion about the topic. Our findings, however, contribute to this knowledge by elucidating increased activities of RiBi during both EMT and mesenchymal-epithelial transition (MET) processes, thereby deepening our understanding of its role. Additionally, we have clarified our novel stance on EMT-targeting strategies. Rather than solely targeting the mesenchymal phenotype, we propose that inhibiting the phenotypic switching ability of tumor cells (a round trip encompassing both EMT and MET) could be more effective, as described in the introduction part.

Additional major revisions:(2) The authors use the FSP1-Cre Model which in the field has been questioned as to not capture all the relevant EMT events and therefore their findings should be corroborated by another EMT model system.

We agree with the reviewer that the Fsp1-Cre model could not capture ALL the relevant EMT events. However, the fidelity and accuracy of Fsp1-Cre model in reporting EMT process of Tri-PyMT cells have also been demonstrated in our previous studies (Lourenco et al. 2020). Also, we have included additional results to further characterize this model: (1) Continuous fluorescence switching from RFP+ to GFP+ was observed in Tri-PyMT cells (Supplementary Fig S1); (2) Bulk RNA-seq data showed the differential expression of EMT marker genes with the RFP+ and GFP+ cells (Supplementary Fig S2A); (3) Single-cell RNA-seq data showed the EMT spectrum and EMT status distributions according to Fsp1(S100a4)/Epcam, and Vim/Krt18 expression (revised Supplementary Fig S3B, 3C). Hope these results clarify the reviewer’s doubt about the Fsp1-Cre model in reporting EMT of tumor cells. Of note, the evaluation of EMT status with RiBi activity does not rely solely on the fluorescent marker switch but on the ETM-related transcriptome (EMTome) of the Tri-PyMT cells.

Again, we agree with the reviewer that the Tri-PyMT model does not report ALL relevant EMT events. In the manuscript, we have included experiments with MD-MB231-LM2 cells (Fig 6D) and analyzed the sequencing databases of breast cancer patients (revised Supplementary Fig S13, S14), to validate the findings of the association between EMT status and RiBi activity.

(3) In the current version of the manuscript, there are no measurements of rRNA synthesis, but the gene expression profiles are used as a proxy for rRNA synthesis. The authors therefore need to include measurements of rRNA synthesis corroborating the RNA sequencing data to support their scientific findings and claims. This can be accomplished by qPCR, Northern blot, or EU staining of the respective sorted cell population. Quantification of rRNA synthesis is also needed for the CX5461/BMH-21 and silencing studies.

We agree that direct measure rRNA synthesis is important to validate the association of RiBi activity with the EMT/MET process. Following the reviewer’s suggestion, we performed EU incorporation assay with RFP+, Double+, and GFP+ Tri-PyMT cells with and without RiBi inhibitors. Under the treatment-naïve condition, the double+ (EMT-transitioning) cells exhibited highest activity of rRNA synthesis compared to either RFP+ (E) and GFP+ (M) cells (revised Supplementary Fig S7). Also, as expected, the treatment of BMH21 or CX-5461 could significantly inhibit the rRNA synthesis (revised Supplementary Fig S8B).

(4) Currently, there is no mechanistic insight as to how rRNA synthesis is increased during EMT, which would also strengthen the manuscript. This could be done through targeted ChIP analysis.

The experimental data in the current manuscript suggest that the activation of RiBi is upstream of the EMT process, as the impaired RiBi pathway hinders the EMT of tumor cells. We are uncertain about the suggestion regarding ChIP analysis. If the reviewer refers to ChIP analysis with EMT transcription factors (i.e., Snail, Twist, and Zeb1), it may not elucidate the mechanisms by which the EMT process is associated with rRNA synthesis. Using sorted GFP/RFP double-positive Tri-PyMT cells, we found enhanced activations in the ERK and mTOR pathways in the EMT-transitioning cells (Figure 3A). It is well-documented that the ERK and mTOR pathways are key coordinators of EMT (Xie et al., Neoplasia 2004; Shin et al., PNAS 2019; Lamouille et al., J. Cell Sci. 2012; Roshan et al., Biochimie 2019). Interestingly, we also observed significantly higher phosphorylation of rpS6, a downstream indicator of mTOR pathway activation, in the Doub+ cells. As an indispensable ribosome protein, rpS6 phosphorylation could impact ribosome functions of protein translation (Bohlen et al., Nucleic Acid Res. 2021; Mieulet et al., 2007).

(5) rRNA synthesis has canonically been linked to the cell cycle therefore it will be necessary for the authors to determine the cell cycle state of their respective cell populations throughout the manuscript.

Following the reviewer's suggestion, we analyzed the cell cycles of RFP+, GFP+, and Doub+ Tri-PyMT cells. Our analysis revealed that the proportion of proliferating RFP+ cells (in the S phase) was higher than that of proliferating GFP+ cells. Interestingly, the Doub+ cells also exhibited a higher ratio of proliferation, which was significantly greater compared to both RFP+ and GFP+ cells (revised supplementary Figure S1B).

(6) Statistics and quantifications are currently missing in several figures and need to be better explained throughout the manuscript to strengthen the scientific rigor of the studies.

We have improved the clarity of our manuscript. Proper statistics descriptions of experiments have been carefully reviewed and adequate information was edited in the revised manuscript.

(7) Only metastasis studies are shown in the current version of the manuscript. These studies should be complemented with primary tumor studies as the main focus of the paper is the contribution of EMT to chemoresistance.

We appreciate the reviewer's suggestion regarding the primary tumor studies. We apologize for not stating clearly in our manuscript. In response, we have revised the manuscript to outline the rationale for establishing a competitive model by injecting a mixture of RFP+ and GFP+ cells in a 1:1 ratio via the tail vein. This model is designed to study of both EMT and MET processes under chemotherapy at a distal site, where tumor cells need phenotypic switches (both EMT and MET) to adapt to and overcome chemo/environmental challenges in this context. Indeed, we have studied the primary tumor growth with the pre-EMT (RFP+) and postEMT (GFP+) cells. Their differential contribution to tumor growth was published in another paper (Lourenco etal. Cancer Res 2020).

**Reviewer #2 (Recommendations For The Authors):**
Figure 1 and associated supplementary figure panelsFig. 1A. More details are needed about the Tri-PyMT model and the induction of EMT in vitro. The authors mention that when growing the isolated cells they spontaneously undergo EMT when grown in 10% FBS. What is the timeline for this transition and how reproducible is it? This information is not clear from Supp. 1. When were cells taken for analysis and also how long is plasticity maintained? According to Supp 1. cell generation 15-21 seems to have a stable cell population of green, red, and yellow cells. Are these cell populations changing if one stimulates the whole cell population with a pro-EMT stimulus? Since cell proliferation is linked to rRNA synthesis the authors also need to include markers of cell cycle for the individual cell population to identify which cell cycle state each sorted cell population is associated with.

We thank the reviewer for recommending further analysis of the cell cycle among RFP+, GFP+, and Doub+ cells. As illustrated in the revised Supplementary Figure 1B, an increased proportion of RFP+ cells was observed in the S phases in comparison to GFP+ cells. Conversely, Doub+ cells demonstrated a proliferation rate even higher than to that of RFP+ cells.

Upon sorting, RFP+ cells were found to spontaneously undergo epithelial-mesenchymal transition (EMT) when cultured in 10% FBS media, thereby converting to GFP+. We quantified the GFP+ cell percentage within the total cell population, noting a consistent transition of a certain proportion of RFP+ cells to EMT, leading to an accumulation of GFP+ cells. This accumulation stabilizes as approximately 60-70% of the entire population become GFP+. Remarkably, re-sorting RFP+ cells from this balanced tumor cell population resulted in a similar fluorescent transition pattern as observed in the parental population. The mechanisms by which tumor cells regulate the EMT phenotypes across the entire population remain unclear. Nevertheless, the equilibrium between RFP+ and GFP+ cells may be attributed in part to the more rapid proliferation of RFP+ cells and the limited proportion of tumor cells undergoing EMT.

We conducted repeated long-term cultures (up to 20 passages) of the Tri-PyMT cells, yielding consistent results. The fluorescence transition pattern in Tri-PyMT cells proved highly reliable. Further details regarding the Tri-PyMT cells have been incorporated into the Methods section.

Fig. 1B. The loading control is not even and quantification is missing, in the text, it states Vimten instead of Vimentin.

The less loading with Doub+ cells was due to the limited number of EMT transitioning cells we could purify by flow sorting. Even though, the expression of both epithelial and mesenchymal markers in the Doub+ cells were clear. In the revised manuscript, we have quantified the Western blot results. We also apologize for the type errors and have corrected the spelling of "Vimentin."

Fig. 1K. In this figure, the authors write: 'It is worth noting that with the 2-phase classifications (Epi or Mes), the elevated RiBi activity was associated with the transitioning cells still exhibiting overall epithelial phenotypes; RiBi activities diminished as cells completed their transition to the mesenchymal phase'. But in Fig. 1K, the Ribi activity is already at a peak during the epithelial state and starts declining already at the beginning of the transition, can the authors please explain this data a bit more? The finding that ribosome biogenesis diminishes once the cells have completed their transition was shown in Prakash et al, Fig. 1 J, I, and accordingly their scientific findings should be discussed in the context of published work.

We acknowledge the reviewer's concerns regarding the comparison of the timeline for EMT in our model with that in Prakash's study. In our model, EMT-transitioning cells are identified by their EMT marker genes and fluorescence expression. We enriched the EMT transitioning cells by sorting the Doub+ cells. Due to the RFP protein's half-life, cells remain RFP+ for 2-3 days after the reporter cassette has switched to GFP expression. In Prakash's study, the EMT transitioning phase was defines by the duration of TGF-β stimulation.

In Figure 1K, cells are categorized based on their EMT pseudotime, calculated from their expression of EMT marker genes in the EMTome. Ribosome biogenesis (RiBi) activity is highest in cells transitioning between phase 1 (Red) and phase 2 (Green), with both phases displaying predominantly epithelial phenotypes (Figures 1C, 1D, and 1E). RiBi activity declines in cells in phases 4, 5, and 3, which exhibit a mesenchymal phenotype. We have expanded the discussion to include more details in comparison with Prakash's study in the revised manuscript.

Supp Fig S4. The authors should provide a rationale for how and why the specific marker genes were selected to calculate the AUC values.

We have chosen the specific EMT marker genes based on their overall expression levels in Tri-PyMT cells, ensuring consistency with the reported associations of their expression patterns to epithelial or mesenchymal phenotypes in the literature. We provide a detailed rationale for the selection of these genes in the Method of revised manuscript (Page #7).

Figure 2 and associated supplementary figure panel. In this figure, rRNA synthesis needs to be evaluated in the cells isolated from the lungs to corroborate the RNA sequencing findings.

Following the reviewer’s suggestion, we performed an RT-PCR of Ribi related genes including Bop1, Gemin4, Its1, Its2, Npm1, Rpl8, Rpl29, Rps9, Rps24, Rps28, Polr1a, Setd4, Utp6, and Xpo1. Consistent with the bulk and single cell RNA sequencing, relatively higher expression of Ribi related genes were detected in Doub+ cells compared to that of RFP+ and GFP+ cells (revised Supplementary Fig S5).

Fig 2C, as per figure Supp Fig S4 please explain the rationale for how and why the specific marker genes were selected.

The same marker genes used for the calculation of the EMT AUC value as in Fig. 1. These marker genes were selected because their overall expression levels are readily detectable in Tri-PyMT cells, their expression patterns are consistent with their epithelial or mesenchymal phenotypes, and the associations between expression of marker genes and phenotypes are in line with the previous reports in literature. Description of AUCell value quantification was included in the revised manuscript (Page #7).

Fig. 2G. The high Ribi during the epithelial state is most likely due to the resumption of cell proliferation of these cells. The authors should check the cell cycle states of these different sets of cells.

We agree with the reviewer that higher Ribi activity could be related to the resumption of cell proliferation of mesenchymal tumor cells. To clarify this, we revisited the scRNAseq data, and project the S phase score to the scatter plot of Ribi activity/MET pseudotime. Indeed, cells in the far mesenchymal state show low S phase score, while the proliferating cells were mostly detected in the MET transitioning phase and epithelial phase (revised Supplementary Figure S6D).

Suppl Fig. 5 Please correct the figure legends as there is no figure D.

We apologize for the mislabeling. We have corrected the figure legend accordingly.

Figure 3. Please explain the rationale for stimulating cells with FBS for the selected time points.Fig. 3A. The loading control is not even, and quantification is missing. In addition, the authors should explain why the different time points were chosen and why FBS was chosen as a stimulus. In addition, from which passage of cells were these cells?

The RFP+ Tri-PyMT cells underwent EMT and switched their expression of fluorescent marker to GFP+ when cultured with FBS. To investigate the response of cells at varying EMT statuses to an FBS-enriched environment, we isolated RFP+, Doub+, and GFP+ cells from the 4th and 5th passages of Tri-PyMT cells and probed downstream signaling pathways after FBS stimuli. The timeline for stimulation was informed by the innate activation profile of these phosphorylation-dependent signals, spanning from 10 minutes to 1 hour. We noted that ERK signaling activation in RFP+ cells occurred within minutes of FBS exposure and diminished within approximately one hour. This ERK signal was more pronounced and persisted longer in Doub+ cells. In contrast, GFP+ cells exhibited a more transient and lower ERK activation (see revised Fig 3A). To address concerns regarding potential uneven loading in our previous assays, we have now included the quantification of Western blots in the revised Fig 3A.

How and why were ERK and mTORC1 pathways chosen for analysis downstream of increased rRNA synthesis? ERK and mTORC1 have mostly been investigated in the role of cell proliferation which is why the cell cycle status of these cell populations will be important to consider in the context of their findings.

The regulation of ribosome biogenesis (RiBi) is mediated by multiple pathways, including the myelocytomatosis oncogene (Myc), mammalian targets of rapamycin (mTOR), and noncoding RNAs, as detailed by Jiao et al. in Signal Transduction and Targeted Therapy (2023). There was no significant difference in Myc expression between tumor cells with epithelial and mesenchymal phenotypes. We thus investigated the activation of the mTOR pathway in sorted RFP+, Doub+, and GFP+ cells. Additionally, given the recognized role of the ERK/MAPK signaling pathway in regulating protein synthesis and cell proliferation, we also analyzed the activation of ERK signals.

In alignment with the reviewer's observation regarding the potential correlation between cell proliferation rate and RiBi activation, we further characterized the cell cycle distributions of RFP+, Doub+, and GFP+ cells. Notably, the Doub+ cells exhibited a higher ratio of cells in the proliferative state (including S and G2/M phases) compared to RFP+ and GFP+ cells. Also, higher percentage of S phase cells were detected in RFP+ cells than GFP+ cells (revised Supplementary Figure S1B).

Figure 3 B, C, D. Please provide more information about which cells are analyzed in this figure.

We apologize for the previous ambiguity regarding the cells analyzed in these figures. To clarify, the figure legend has been revised to specify that Tri-PyMT cells from the 5th to 10th passages were the subjects of analysis for cell size and nascent protein synthesis, utilizing flow cytometry.

Figure 3D. The selected images show enlarged nucleoli/ fibrillarin which is an indicator of increased rRNA synthesis however, the authors need to show an increase in rRNA transcripts by q-PCR or Northern blot and also show EU staining in these different cell states to support their claim.

We appreciate the reviewer's recommendation to further validate the enhanced ribosome biogenesis (RiBi) in Doub+ cells. In response, we conducted RT-PCR analysis of several RiBi-related genes (revised Supplementary Fig S5). Additionally, we carried out an EU incorporation assay to illustrate the rRNA transcription activity within these cells. The new results have been incorporated into the revised manuscript (Supplementary Fig S7).

Figure 4 and associated supplementary. In this figure, the authors show that using small molecule Pol I assembly inhibitors (BMH-21 and CX-5461) reduces the expression of mesenchymal proteins. As mentioned in previous comments these results should be put in the context of published work by Prakash et al which demonstrate that upon CX-5461 and genetic silencing of Pol I EMT is hampered as demonstrated by gene expression profiles as well as functional assays.

We revised the description of our experiments with Pol I inhibitors in the revised manuscript by including the citation context (Prakash et al Nat Commun, 2019) as mentioned above.

Figure 4A. Please provide an explanation of how the doses of Pol I assembly inhibitors were determined and also the selected time points. The Pol I assembly inhibitors should have an effect within a few hours (Drygin, Cancer Research, 2011, Peltonen, Cancer Cell, 24). The authors also need to show that the BMH-21 and CX5461 at selected doses are indeed inhibiting rRNA synthesis in the selected cell populations. The data would also be strengthened by performing ChIP analysis demonstrating that indeed the Pol I complex is disassociated from the rDNA genes upon inhibition.In addition, why are there only 2 reports and how were the statistics done? Were the data normalized to the total number of cells? The graph visually shows a difference in cell numbers. Are cells dying at this concentration? More controls must be included including markers for cell stress, p53, autophagy, and apoptosis.

The dose of Pol inhibitors was selected based on prior studies, as noted by the reviewer. Peltonen et al. demonstrated that BMH-21 inhibits growth across a wide spectrum of cancer cell lines, achieving a mean half-maximal inhibition of cell proliferation (GI50) at 160 nM (Peltonen K., et al. Cancer Cell. 2014). Consistently, in our experiments, the growth inhibitory effect of BMH-21 on Tri-PyMT cells fell within this range, at approximately 200 nM (Fig 5B, Supplementary Fig S10).

To address the reviewer's suggestion and verify that RiBi inhibitor effectively inhibits rRNA synthesis in our study, we conducted an EU incorporation assay. This assay revealed significant inhibition of rRNA synthesis by BMH-21 and CX5461 in Tri-PyMT cells (revised Supplementary Fig S8B). Furthermore, to enhance the robustness of our findings, we repeated the BMH-21 treatment on sorted RFP+ Tri-PyMT cells across three biological replicates, which yielded consistent results.

Figure 4B. How many replicates were done for this experiment and please provide quantification as per previous comments on WB experiments. The authors should provide a rationale for why Snail and Vimentin were chosen for these studies. Also, the authors should provide a functional assay and demonstrate that cells are less migratory post-treatment and not only markers.

Western blots with sorted Tri-PyMT cells were performed twice. We have added the quantification of these blot in the revised manuscript. Snail and Vimentin were chosen as mesenchymal markers to indicate EMT phenotype switches as those were well-studied and commonly used mesenchymal markers of EMT. The association of fluorescent marker switch and

EMT phenotype such as cell migration was well established in our previous study (Fischer et al., 2015, Lourenco et al., 2020). The morphology and migration property of GFP+ were well distinguished from RFP+ counterparts. Also, following reviewer’s suggestion, we performed migration assay with BMH21 treatment (revised Supplementary Fig 8C). Indeed, the treatment with BMH21 or CX5461 inhibited cell migration as expected.

Supplementary figure 7. The authors need to provide a rationale as to why the two Rps were chosen to inhibit ribosome biogenesis.

The two Rps targets were chosen based on their differential expression in Doub+ cells compared with RFP+ and GFP+ cells. Also, we considered the overall expression level of these genes in Tri-PyMT cells. We have edited the according text in the revised manuscript.

Figure S7B. In the images shown there does not appear to be a significant change in the number of nucleoli however the cells seem to be smaller. This should be explained.

We agree with the reviewer that the box plot does not clearly show the nucleoli differences between these cells. We present the data with a violin plot, which more clearly exhibit the result (revised Supplementary Fig S9B). It was also true that the sizes of the Rps knockdown cells were relatively smaller than control cells. This is consistent with the finding that the EMT transitioning cell size was bigger than the non-transitioning cells (Fig 3B)

.

Figure 5 and Supp 8. The authors should provide the background as to why the specific chemotherapeutic drugs were chosen.

The chemotherapeutic agents employed in this study are widely used in the treatment of breast cancer. For instance, Cyclophosphamide (CTX) hampers both DNA replication and RNA transcription; Doxorubicin inhibits DNA replication by disrupting topoisomerase activity; Paclitaxel prevents cell division by stabilizing microtubules; and 5-Fluorouracil (5-FU), a pyrimidine analog, blocks thymidylate synthase, thereby disrupting DNA synthesis. Additionally, some of these agents, such as CTX and 5-FU, may directly or indirectly affect RNA polymerase, prompting us to investigate the synergistic effects of these drugs when used in combination with BMH21. We have included the information in revised manuscript.

Fig 5B/Supp 8. Can the authors please explain why only 2 replicates were done and provide a rationale for future statistics?

Using serial concentrations of drugs tested—6 doses for BMH21 and 8 doses for CTX—it is logical to arrange the experiment in duplicates on 96-well plates. For the statistical analysis, we conducted dose-response analysis to ascertain the IC50 values for each drug alone and in combination. Additionally, we calculated the synergy score to assess the interactions between the drugs. The methodology section of the manuscript has been enhanced to provide a clearer description of these processes in the revised version.

Figure 6. The authors should provide a rationale of why tail veins were chosen as their in vivo model system as the EMT cells do not cause metastasis and if chemoresistance is the main focus of their studies both primary and secondary tumors should be considered. Why was not the MMTVPyMT mouse model chosen where the cells were originally isolated from to test the role of the dual treatment? How was the drug concentration decided and the interval of treatments?

We acknowledge the reviewer's concerns regarding the choice of experimental setup for our metastasis model. Certainly, utilizing the original MMTV-PyMT mice for the combination therapy experiment would be the ideal scenario. However, there are potential drawbacks to using these transgenic mice: (1) The occurrence of multiple primary tumors that develop simultaneously but without synchronized timelines (in mice aged 6-9 weeks), and the unsynchronized development of lung metastasis (from 10-16 weeks of age). This leads to uncontrollable variations in the experimental setup, particularly when establishing multiple treatment groups; (2) Gathering a sufficient number of female transgenic mice of a similar age poses another challenge; (3) The absence of tumor cell labeling complicates the focus on assays for EMT/MET phenotype changes during tumor progression. Consequently, we have chosen to employ our Tri-PyMT model for this experiment. The drug treatment protocol was established after reviewing literature on the in vivo application of CTX and BMH21 treatment (Peltonen etal. Cancer Cell 2014; Jacobs etal. JBC 2022).

Figure 6B, C. The authors should provide quantification for these data, how many mice were analyzed, and how many sections were stained and analyzed.

We have improved the quality of these fluorescent images and clarify the methodology, including the mouse/section numbers per group, for obtaining these fluorescent images in the legend. To quantify the differential impact of BMH21 on RFP+ and GFP+ tumor cells, we performed flow cytometry (revised Supplementary Fig S11). We have also changed the presentation of these flow data to improve the clarity of these results.

Fig 6D. How were the treatment timeline and dosing chosen? LM2 cells are derived from a metastatic site, so they are not transitioning cells they are stably mesenchymal why was this chosen as their in vivo model?

LM2 cells were derived from the lung metastasis of MDA-MB-231 cell line. These cells exhibit predominantly mesenchymal phenotype in culture. While growing into metastasis in the lung, expressions of epithelial markers such as E-cad were upregulated (Supplementary Fig S12), suggesting a MET process may be involved the outgrowth of lung metastasis. Therefore, we choose the LM2 cells as our experimental model for assessing the effect of RiBi inhibitor on MET. The treatment timeline was determined based on previous studies of BMH21 and chemotherapy applications in vivo (Peltonen etal. Cancer Cell 2014; Jacobs etal. JBC 2022).

**Reviewer #3 (Public Review):**
Summary:Ban et al. investigated the role of ribosome biogenesis (RiBi) in epithelial-to-mesenchymal transition (EMT) and its contribution to chemoresistance in breast cancer. They used a Tri-PyMT EMT lineage-tracing model and scRNA-seq to analyze EMT status and found that RiBi was elevated during both EMT and mesenchymal-to-epithelial transition (MET) of cancer cells. They further revealed that nascent protein synthesis mediated by ERK and mTOR signaling pathways was essential for the completion of RiBi. Inhibiting excessive RiBi impaired EMT and MET capability. More importantly, combinatorial treatment with RiBi inhibitors and chemotherapy drugs reduced metastatic outgrowth of both epithelial and mesenchymal tumor cells. These results suggest that targeting the RiBi pathway may be an effective strategy for treating advanced breast cancer with EMT-related chemoresistance.Strengths:The conclusions of this study are generally supported by the data. However, some weaknesses still exist as mentioned below.Weaknesses:(1) The study predominantly focused on RiBi as a target for overcoming EMT-related chemoresistance. Thus, it will be necessary to provide some canonical outcomes after upregulating ribosome biogenesis, such as translation activity. I would suggest ribosome profiling or puromycin-incorporation assay, or other more suitable experiments.

EU incorporation assay (revised Supplementary Fig S7) and puromycin incorporation assay (Fig 3C) were performed.

(2) The results were basically obtained from mice and in vitro experiments. While these results provide valuable insights, it will be valuable to validate part of the findings using some tissue samples from patients (e.g. RiBi activity) to determine the clinical relevance and potential therapeutic applications.

We agree. We have added the analyses on the correlation between patients’ survival and RiBi activation (revised Supplementary Fig S13, S14).

(3) The results revealed that mTORC1 and ERK mediated RiBi activation. How about mTORC2? It will be informative to evaluate mTORC2 signaling.

We investigated the role of the mTORC1 pathway in regulating RiBi activation. It is pertinent to acknowledge that the mTORC1 complex is known to positively regulate protein synthesis through the phosphorylation of ribosomal protein S6 kinase, among other mechanisms. Additionally, Rps6 is recognized as an essential component of the 40S subunit in the ribosome. We agree with the reviewer that mTORC2 may also be involved in RiBi activity, as its activation is mediated through ribosome association (Zinzalla et al., Cell 2011; Prakash et al., Nat Comm 2019). However, this association is more likely to be downstream of RiBi activation, as the RiBi inhibitor CX5461 can block the translocation of Rictor into the nucleus (Prakash et al., Nat Comm 2019).

We also revisited our sequencing data of RFP+, GFP+, and Doub+ cells. While there was no significant change in the expression of either Rptor or Rictor among these cells, the LSMean (overall expression level) of Rptor was higher than that of Rictor; for example, 163.77 vs 29.95 in RFP+ cells. This suggests that mTORC1 may play a dominant role in regulating RiBi activity in our model.

Furthermore, we analyzed how Rapamycin (an mTORC1 inhibitor) affects the EMT process in TriPyMT cells. As expected, Rapamycin-treated cells exhibited higher expression of the epithelial marker E-cadherin (Ecad) and lower expression of the mesenchymal markers Snail and Vimentin (Vim) compared to the control (For Reviewers Figure 3).

(4) The results also demonstrated promising synergic effects of Pol I inhibitor (BMH21) and chemotherapy drug (CTX) on chemo-resistant metastasis. How about using the inhibitors of mTORC1 together with CTX?

Several mTOR inhibitors (e.g., sirolimus, temsirolimus, ridaforolimus) have demonstrated antitumor activity. The combination of mTOR inhibitors with various targeted therapies or chemotherapies is being examined in numerous clinical trials, showing promising results. Although the combination therapy of mTORC inhibitors and CTX is beyond the scope of our study, we analyzed how mTOR inhibitors may affect the EMT process in our model, as mentioned above. Western blot analysis of EMT markers (E-cadherin, Snail, and Vimentin) showed that rapamycin treatment inhibited the EMT transition of Tri-PyMT cells. (For Reviewers Figure 3).

(5) While the results demonstrate the potential efficacy of RiBi inhibitors in reducing metastatic outgrowth, other factors and mechanisms contributing to chemoresistance may exist and need further investigation. I would suggest some discussion about this aspect.

Following reviewer’s suggestion, we have edited the discussion section with more future directions.

**Reviewer #3 (Recommendations For The Authors):**
(1) Please provide the quantified data for all western blots, rather than solely show some representative blots.

We quantified the western blot images as shown in the revised figures. Thanks for reviewer’s suggestion.

(2) Please add a graphic abstract or schematic to help the readers understand the whole story.

We have summarized a schematic graph of our findings in the revised manuscript (Supplementary Fig S15).

(3) It is hard to read the numbers inside all plots of flow cytometry.

High-resolution figures of flow plots are included in the revised manuscript.

(4) Please provide high-resolution figures for all the synergy plots.

High-resolution figures of synergy plots are included in the revised manuscript.